# An injury-induced serotonergic neuron subpopulation contributes to axon regrowth and function restoration after spinal cord injury in zebrafish

Chun-Xiao Huang[1,2], Yacong Zhao[1,2], Jie Mao[1,2], Zhen Wang[1,2], Lulu Xu[1,2], Jianwei Cheng[1,2], Na N. Guan [1,2,3,4 ✉] & Jianren Song [1,2,3,4 ✉]

Spinal cord injury (SCI) interrupts long-projecting descending spinal neurons and disrupts the spinal central pattern generator (CPG) that controls locomotion. The intrinsic mechanisms underlying re-wiring of spinal neural circuits and recovery of locomotion after SCI are unclear. Zebrafish shows axonal regeneration and functional recovery after SCI making it a robust model to study mechanisms of regeneration. Here, we use a two-cut SCI model to investigate whether recovery of locomotion can occur independently of supraspinal connections. Using this injury model, we show that injury induces the localization of a specialized group of intraspinal serotonergic neurons (ISNs), with distinctive molecular and cellular properties, at the injury site. This subpopulation of ISNs have hyperactive terminal varicosities constantly releasing serotonin activating 5-HT$_{1B}$ receptors, resulting in axonal regrowth of spinal interneurons. Axon regrowth of excitatory interneurons is more pronounced compared to inhibitory interneurons. Knock-out of *htr1b* prevents axon regrowth of spinal excitatory interneurons, negatively affecting coordination of rostral-caudal body movements and restoration of locomotor function. On the other hand, treatment with 5-HT$_{1B}$ receptor agonizts promotes functional recovery following SCI. In summary, our data show an intraspinal mechanism where a subpopulation of ISNs stimulates axonal regrowth resulting in improved recovery of locomotor functions following SCI in zebrafish.

[1] Translational Research Institute of Brain and Brain-Like Intelligence, Shanghai Fourth People's Hospital, School of Medicine, Tongji University, Shanghai 200434, China. [2] Department of Anatomy, Histology and Embryology, School of Medicine, Tongji University, Shanghai 200092, China. [3] Clinical Center for Brain and Spinal Cord Research, Tongji University, 200092 Shanghai, China. [4] These authors contributed equally: Na N. Guan, Jianren Song. ✉email: naguan@tongji.edu.cn; song.jianren@tongji.edu.cn

Locomotor behaviors are generated by descending motor commands from brain locomotor regions activating a spinal central pattern generator (CPG)[1–3]. The long axons of neurons from the descending brain locomotor nuclei synapse with spinal local neurons and are considered to be the primary excitatory drive of the spinal CPG[4–6]. Appropriate locomotor parameters are determined by the interplay of synaptic connections between excitatory and inhibitory interneurons in spinal CPG[1,2,7,8]. Recruitment of spinal glutamatergic interneurons, which project over many spinal segments, can generate locomotion[8–11], set locomotor patterns[12–15] and coordinate the body's rostral–caudal movement[5,16]. Spinal cord injury (SCI) causes locomotor dysfunction by severing the supraspinal connections as well as disrupting the neuronal network in spinal CPG[17,18]. Restoration of accurate locomotion requires reestablishment of neural circuits in spinal CPG[19]. The paucity of regrowth of injured axons in the adult mammalian central nervous system (CNS) is a major hurdle in understanding the intrinsic mechanisms that promote appropriate axon regrowth. The ready regeneration of adult zebrafish CNS[20–22] and repair of spinal circuits[23] after injury allows one to explore the intrinsic mechanisms governing axon regrowth of spinal CPG interneurons and the reconnection of local CPGs after spinal transection.

In vertebrates, serotonin is a critical neurotransmitter synthesized and released by serotonergic neurons in the ventral raphe nucleus[24] and spinal cord[25] that modulates intrinsic properties of neurons[26,27], synaptic plasticity[28,29], and locomotor outputs[26,30]. Several lines of evidence suggest that spinal serotonergic signals promote the recovery of locomotion after SCI[27,31–33]. Increased descending serotonergic innervation rostral to the injury site has been observed and regrowth of serotonergic fibers caudal to the site of injury correlated well with recovery of locomotor function after SCI[31,34,35]. In zebrafish, marked injury-induced regeneration of intraspinal serotonergic neurons (ISNs) around the lesion sites that occurred via shh signaling and preceded recovery of locomotor function has been reported[34]. The emergence of ISNs has also been reported in the turtle after SCI[36]. While the molecular and cellular diversity of serotonergic neurons under physiological conditions is well-known[37], it is not clear whether the regenerated ISNs form a specialized subpopulation that is preferentially activated to initiate and maintain the repair and recovery process. Administration of exogenous serotonin or serotonin reuptake inhibitors facilitated recovery of locomotor function after SCI by promoting motor neuron regeneration in zebrafish[38] and modulating intrinsic electrophysiological properties of neurons in mammals[29,33,39]. The complexity of 5-HT receptors combined with the lack of selective drugs have hindered our understanding of how serotonin promotes axon regrowth and spinal neural circuit reestablishment after spinal injuries. Recent advances in the field of genetic engineering make genetic manipulation of 5-HT receptor subtypes a more effective approach to identify their specific roles in promoting axon regrowth of different CPG interneurons after SCI.

In the present study, we used zebrafish to investigate the neural intrinsic mechanisms promoting axon regrowth of spinal CPG interneurons after spinal cord transection. We identified a specialized subpopulation of injury-induced ISNs that congregated around and along the injury site. Chemogenetic ablation of this subpopulation in a two-cut injury model, which removed brain-descending influences, revealed that this specialized subpopulation was indispensable for promoting axon regrowth of spinal interneurons and restoration of locomotor function. Using a combination of electrophysiological, functional imaging, liquid chromatography–mass spectrometry (LC–MS) and RNA sequencing techniques, we found that the injury-induced ISNs possess distinct intrinsic properties, release serotonin constantly from hyperactive terminal varicosities and express unique transcription factors compared with ISNs located in distant spinal segments. We confirmed by knockout of different 5-HT receptor subtypes and pharmacological agents that these injury-induced ISNs specifically activate 5-HT$_{1B}$ receptors and thus promote axon regrowth of glutamatergic interneurons in spinal CPG. The whole event results in reorganization of spinal circuit after SCI. In summary, we describe an injury-induced subpopulation of ISNs releasing serotonin that promote axon regrowth of spinal excitatory interneurons and reestablishment of spinal CPG by acting specifically on the 5-HT$_{1B}$ receptor subtype. The serotonergic subpopulation may be amenable to cell transplantation therapy and the 5-HT$_{1B}$ receptor is a good candidate target for treatment after SCI.

## Results

**Aggregation of regenerated ISNs positively correlated with axon regrowth and recovery of locomotor function after SCI.** In juvenile/adult zebrafish, the completely transected spinal segments regenerated, gradually joined up and had recovered to the size of a normal spine after about 8 weeks post injury[34] (Fig. 1a, b; Supplementary Fig. 1a, d). First, we characterized the dynamic changes in the number of ISNs at the injury site during the recovery process using SCI animals of the *Tg(tph2: GFP)* line, in which more than 97% GFP$^+$ neurons were labeled by anti-serotonin antibodies (Supplementary Fig. 1b)[40]. In zebrafish, the ISNs population reaches morphological and functional maturity at 4 days post fertilization[25,41] and in the current study, the soma number and distribution of ISNs stayed constant up to four months of age (Supplementary Fig. 1c). We found congregation of the soma of ISNs towards and along the margins of regenerating spinal segments in response to injury (Fig. 1b). The number of ISNs at the injury sites decreased 1 week post injury (Supplementary Fig. 1d), then increased dramatically and reached a plateau 4 weeks post injury (Fig. 1b). We next evaluated locomotor function. Free-swimming speed and maximum speed of forced-swimming recovered gradually in these SCI animals being 50% at 6 weeks and more than 70% at 8 weeks post injury (Fig. 1c, d). Electrophysiological recordings showed that restoration of locomotor function was accompanied by recovery of the normal time lag of rostro–caudal (R–C) signal propagation, which is a measure of the regrowth of severed axons of spinal interneurons across the injury site into caudal segments (Fig. 1e). In the uninjured animals, the R–C time lag of locomotor bursts between L4 and L25 spinal segments decreased with increased swimming frequency (Fig. 1e), showing the appropriate signal propagation for R–C body coordination (Fig. 1f gray dots)[10,42]. At 1 week post injury, when the transected spinal segments were still separate (Supplementary Fig. 1a, d, f), no locomotor signal propagation was recorded in segments caudal to the injury sites (Fig. 1e). By 2 weeks post injury, the SCI animals could generate R–C body coordination and locomotor patterns (Fig. 1e). The time lag was more pronounced in zebrafish at 2 weeks post injury (Fig. 1f blue dots), and was similar to that in uninjured fish at 8 weeks post injury (Fig. 1f purple dots), indicating substantial axon regrowth of spinal interneurons across the injury site into the caudal spinal segments and reestablishment of spinal CPG during the recovery process[2,7,10,43,44]. We quantified the extent of axon-regrown spinal interneurons along the recovery time by retrograde injection of rhodamine dextran (RD) into the spinal cord two segments caudal to the injury site (Fig. 1g, Supplementary Fig. 1e) in the expectation that if the severed axon of interneurons regrew across the injury sites, the soma of these neurons would be retrogradely labeled. When the transected

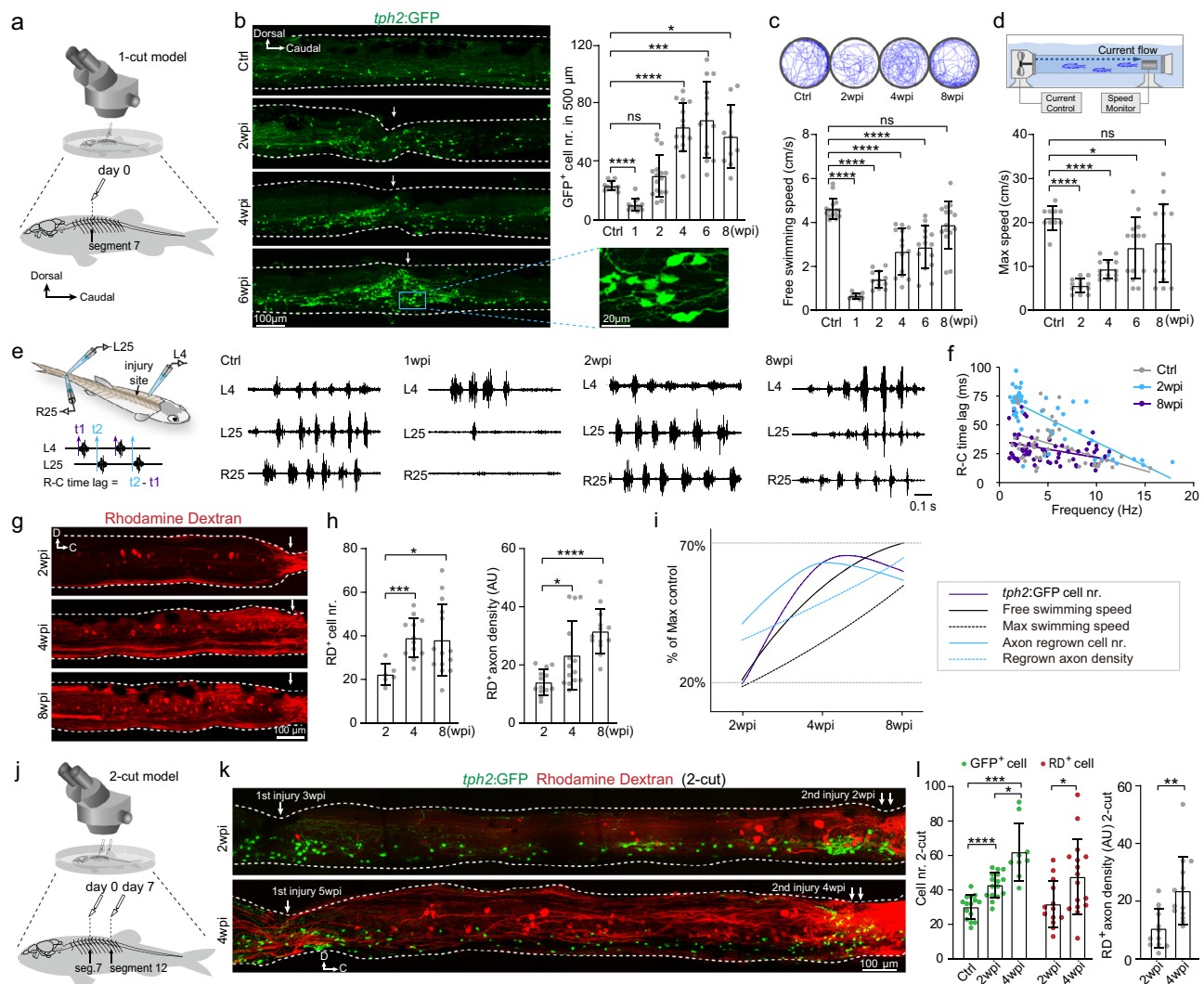

**Fig. 1 Injury-induced regeneration of intraspinal serotonergic neurons (ISNs) is positively correlated with locomotion recovery. a** Illustration of adult zebrafish one-cut spinal cord injury (SCI) model with a complete transection at the seventh segment. **b** (left) Immunohistochemistry images show GFP+ ISNs distribution in the lateral view of whole-mount spinal cord in uninjured (Ctrl) and SCI fish. Injury sites are indicated by white arrows. The GFP+ ISNs within the injury site in the blue box is enlarged. (right) Quantification of GFP+ ISNs numbers in the 500 μm long spinal cord covering the injury site. **c** Five minutes free-swimming traces recorded by camera in the 9 mm open dish, and quantification of free-swimming speed of uninjured and SCI animals. **d** Illustration of a forced-swimming setup, and quantification of maximum water velocity at which uninjured and SCI fish reached exhaustion. **e** Illustration of the multi-channel electromyogram (EMG). Typical EMG traces recorded from uninjured and SCI animals for the same duration were shown. **f** Plots of R–C time lag versus swimming frequency for each swimming cycle in uninjured and SCI animals. $N = 4$ fish in each group. **g** Images show rhodamine dextran (RD) retrograde labeling of spinal interneurons with axon regrowing over injury site after SCI. **h** Quantification of retrogradely labeled spinal interneuron numbers in the 2–4 segments rostral to the injury site and regrown axon density for semispinal thickness laterally. **i** Correlation analysis shows the relationship of GFP+ ISNs numbers in the injury site, the free and maximum swimming speeds, axon-regrown spinal interneuron numbers, and regrown axon density after SCI. y-Axis represent the percentage of maximum value. **j** Illustration of adult zebrafish 2-cut SCI model with a second complete transection at the 12th segment. **k** Immunohistochemistry images show GFP+ ISNs distribution and RD retrograde labeling in two-cut SCI model. **l** Quantification of GFP+ ISNs numbers at the second injury site, RD+ retrogradely labeled axon-regrown spinal interneuron numbers and regrown axon density rostral to the secondary injury site. All data are presented as mean ± SD. *P < 0.05, **P < 0.01, ***P < 0.001, ****P < 0.0001, significant difference. For detailed statistics, see Supplementary Table 1.

spinal tissues were still separated, no retrograde labeling was observed (Supplementary Fig. 1f). Both the number of retrogradely labeled spinal interneurons and density of regrown axons at the injury site increased significantly during the period from 2 to 8 weeks post injury (Fig. 1h). Taken together, these data suggest that after SCI the restoration of well-coordinated locomotor activity relies on reconnection of two disconnected spinal CPGs by regrowth of axons arising from rostrally located spinal interneurons. Our data show that the dynamic increase in injury-induced ISNs correlated with the number of spinal interneurons

that regrew across the injury site, their density and the restoration of locomotor function during recovery (Fig. 1i).

After SCI, the supraspinal serotonergic projections are removed. Limited regrowth of descending serotonergic axons and density of terminal varicosities after SCI have been reported previously in zebrafish[34], indicating that prelesion patterns of innervation are not easily restored. To clarify whether the injury-induced ISNs could independently promote locomotion recovery in the absence of descending serotonergic modulation, we used a two-cut SCI model by inducing a second complete transection

caudally remote from the previous injury site on an established (one-cut) SCI model to evaluate the effect of ISNs at the second injury site (Supplementary Fig. 1g). There was a decrease of more than 90% in serotonergic axon density at 1 week post injury in the region five segments caudal to the injury site (segment 12) of one-cut SCI animals. In the same animals, axon density at two segments rostral to the injury site (segment 5) was similar to that at the uninjured level (Supplementary Fig. 1h). Hence, the second spinal transection was performed at 1 week post SCI at the region five segments caudal to the initial injury site in 1-cut SCI animals (Fig. 1j, Supplementary Fig. 1g). In the 2-cut SCI model, the soma of ISNs also congregated around and along the margin of the second injury site and their numbers were similar to those found at the wound site in 1-cut SCI animals (Fig. 1b, k, l), suggesting that the generation of ISNs is a pure intraspinal response to SCI. To investigate whether these injury-induced ISNs in 1- and 2-cut animals are newly regenerated, we injected EdU intraperitoneally in SCI animals of Tg(tph2:GFP) line daily from 1 to 7 days post injury (Supplementary Fig. 1i). This labeled $41.5 \pm 8.7$ and $15.6 \pm 6.2$ GFP$^+$/EdU$^+$ neurons at the injury site 2 weeks post injury in the 1-cut and 2-cut model, respectively. Double-labeled neurons were significantly fewer in distal spinal segments and were undetectable in spinal segments of uninjured animals (Supplementary Fig. 1i), suggesting a spatial confinement of reactive ISNs regeneration. These findings agree with the results of previous studies, which showed that blocking the serotonin signal did not affect regeneration of ISNs[34,38]. Importantly, associated with the increase of injury-induced ISNs, injection of retrograde tracer revealed that spinal interneurons with axon regrowth across the second injury site were also significantly increased from 2 to 4 weeks post injury (Fig. 1k, l) in the 2-cut SCI animals which precludes descending serotonergic innervation. Locomotion also recovered from 2 to 4 weeks post injury. Maximum swim speed increased from $7.00 \pm 4.69$ cm/s ($n = 10$) to $9.75 \pm 3.53$ cm/s ($n = 5$) ($P = 0.089$, unpaired $t$ test) and free swim speed increased from $1.68 \pm 1.51$ cm/s ($n = 5$) to $2.28 \pm 1.19$ cm/s ($n = 10$) ($P = 0.058$, unpaired $t$ test). Thus, we found that the injury-induced regeneration of ISNs is positively associated with axon regrowth of long-projecting spinal interneurons and reconnection of transected neural circuits. This modulation of spinal interneuron axon regrowth was achieved predominantly through intraspinal mechanisms without the contribution from descending serotonergic axons.

**Chemogenetic ablation of injury-induced ISNs impaired axon regrowth and locomotion recovery.** To determine whether the injury-induced ISNs were indispensable for axon regrowth and recovery of locomotor function after SCI, we used a Tg(tph2:Gal4;UAS:nfsB-mCherry) transgenic line in which the mCherry$^+$ serotonergic neurons could be specifically ablated by metronidazole (MTZ) treatment. In this transgenic line, mCherry labeled about ~50% of the spinal serotonergic neurons and 98% of all mCherry$^+$ neurons contained serotonin (Supplementary Fig. 1j). MTZ treatment (Fig. 2a) significantly reduced mCherry$^+$ ISNs in the injury site at 2 and 4 weeks post injury (Fig. 2b, c). Compared to Tg(tph2:Gal4;UAS:nfsB-mCherry) SCI fish without MTZ treatment, we found a greatly reduced numbers of axon-regrown spinal interneurons (Fig. 2f) and density of regrown axons (Fig. 2g) at injury sites as well as a significantly impaired recovery of locomotor function in fish treated with MTZ (Fig. 2d, e). MTZ injection was reported to increase the production of serotonin in the rat brain[45]. To investigate the effect of MTZ alone, we used a Tg(tph2:Gal4;UAS:mCherry) line as control, in which MTZ treatment will not cause cell ablation. The results showed that axon regrowth and locomotion recovery in Tg(tph2:Gal4;UAS:mCherry)

SCI fish were unaffected by MTZ treatment (Supplementary Fig. 1k). To specifically demonstrate that the observed effect was not induced by global ablation of descending serotonergic projections, we also did the MTZ treatment in the 2-cut SCI model (Fig. 1j, Supplementary Fig. 1g). ISNs at the second injury site were ablated by MTZ treatment (Fig. 2h, i) which impaired both locomotion recovery (Fig. 2j, k) and axon regrowth (Fig. 2l, m) in the 2-cut SCI fish. These results confirmed that the injury-induced ISNs can independently promote axon regrowth of long-projecting spinal interneurons, circuitry reestablishment and locomotor restoration in the absence of descending serotonergic modulation.

**Injury-induced ISNs constitute a molecularly distinct subpopulation constantly releasing serotonin.** We performed whole-cell patch-clamp recording in GFP$^+$ neurons of the Tg(tph2: GFP) line to examine neuronal properties of injury-induced ISNs (Supplementary Fig. 2a). Suprathreshold positive current injection into ISNs at the injury segment triggered action potentials, confirming that they were mature neurons (Supplementary Fig. 2b, c) albeit with highly diverse morphologies (Supplementary Fig. 2d). To further investigate if the regenerated ISNs in the injury segment were functionally different to those ISNs located in the distal segments, we made extracellular recording of the spike activity of the two ISN populations. In the SCI animals at the injured segment level, ISNs displayed spontaneous TTX-sensitive action potentials (Fig. 3a, b) with a mean spike frequency of $5.74 \pm 0.98$ Hz (Fig. 3c, d purple) at 6–8 weeks post injury, while the spontaneous spike activity of those ones in distal segments of the same animals was much lower being $1.56 \pm 0.34$ Hz (Fig. 3c, d green). Next, we used two-photon confocal microscopy to examine the Ca$^{2+}$ homeostasis in the soma and terminal varicosities of ISNs at both the level of the SCI and segments distal to the injury site in SCI animals of the Tg(tph2:Gal4;UAS:GCaMP6) line (Fig. 3e). Ca$^{2+}$ oscillations in the injury-induced ISNs soma (Fig. 3f upper panel; Supplementary Fig. 2e) and varicosities (Fig. 3g upper panel; Supplementary Fig. 2f) displayed significantly higher frequency and larger amplitude than those in distal spinal segments of either injured animals (Fig. 3f–h) or the corresponding segments of non-injured animals (Supplementary Fig. 2g). Furthermore, these data suggest that the injury-induced ISNs are a subpopulation that is absent in the distal spinal segments of SCI animals or spinal segments of uninjured animals. The hyperactivity of Ca$^{2+}$ oscillations in terminal varicosities of injury-induced ISNs suggests that these neurons constantly release serotonin from the terminals. Together, the electrophysiological and Ca$^{2+}$ imaging data indicate that the ISNs regenerated at injury site and those located away from the injury site are functionally distinct subpopulations.

We characterized the neurochemical profile of ISNs in the injured and distal segments. First, we used LC–MS analysis to determine if hyperactive regenerated ISNs at the SCI site released more serotonin than their counterparts in distal segments away from the injury site. The released serotonin concentration measured in the perfusate from the injury segments was significantly higher than in that from the distal segments (Fig. 3i). Second, staining with anti-Glutamate, anti-GABA and anti-ChAT in Tg(tph2:GFP) fish at 4 weeks post injury showed that at the injury segment, about 6% tph2:GFP$^+$ neurons were positive for anti-Glutamate, 6% tph2:GFP$^+$ neurons were positive for anti-GABA, and almost none of these tph2:GFP$^+$ neurons stained for anti-ChAT (Supplementary Fig. 2h–j). These data suggest that serotonin is the predominant neurotransmitter released by regenerated ISNs having hyperactive electrophysiological properties and facilitates axonal regrowth and circuit reestablishment.

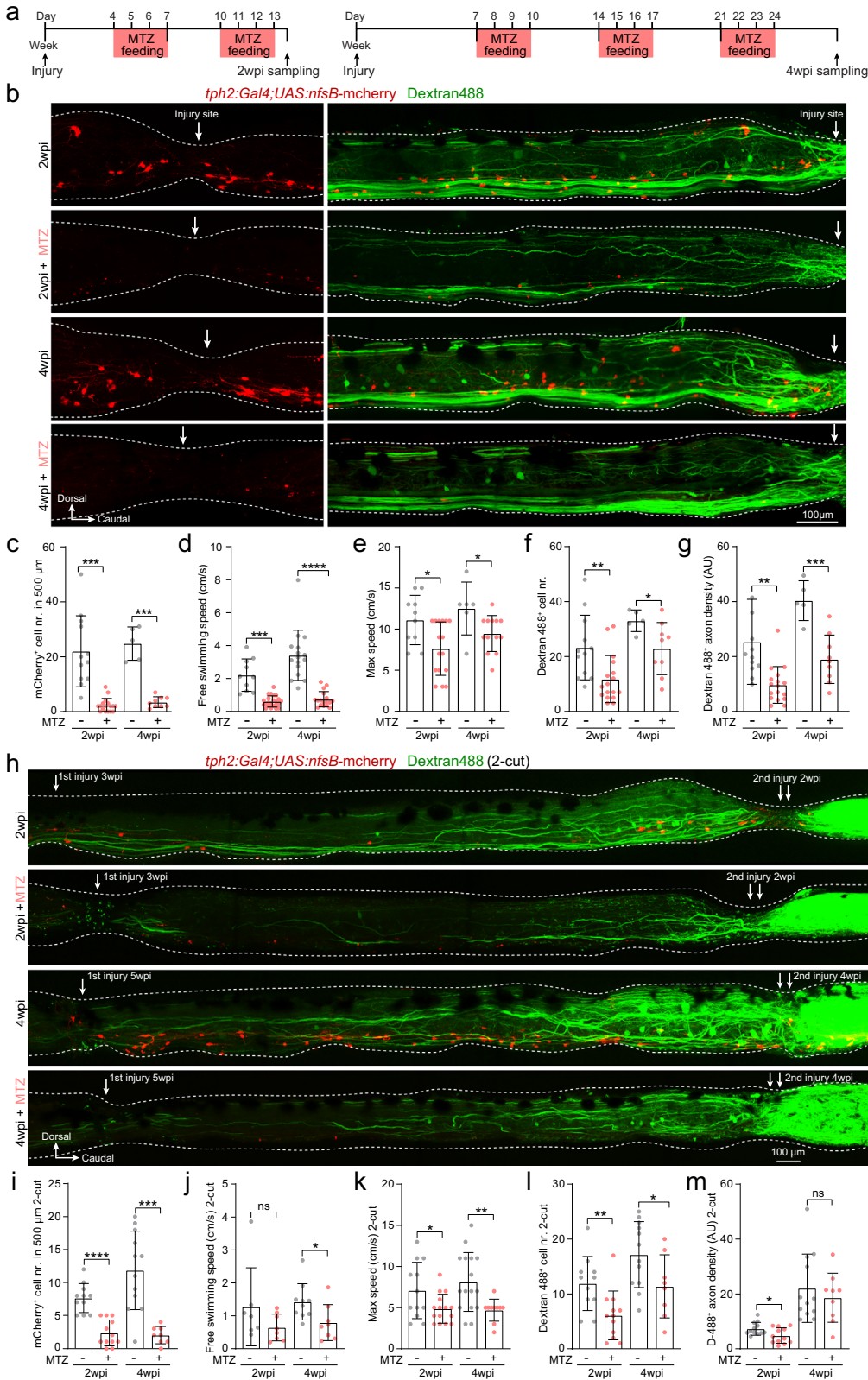

Interestingly, we observed that the injury-induced ISNs project massive axons enwrapping and intertwining with axon of spinal interneurons labeled by RD (Supplementary Fig. 2k, l). This spatial arrangement can be speculated to maintain an elevated concentration of released serotonin around the regrowing axon of spinal interneurons at the injury sites.

The functionally distinct subpopulation of injury-induced ISNs was also embodied by their molecular signature. We analyzed the transcriptome profiles based on RNA-seq data of FAC-sorted ISNs, isolated from either the injury segment (Fig. 3j purple) or the distal segments (Fig. 3j green) of the same preparations at 6 to 8 weeks post injury. The two subpopulations of ISNs shared common serotonergic ID genes for synthesis and transport

**Fig. 2 Injury-induced ISNs are indispensable for axon regrowth and locomotion recovery. a** Metronidazole (MTZ) treatment protocols used to genetically ablate mCherry[+] ISNs after SCI using *Tg(tph2:Gal4;UAS:nfsB-mCherry)* line. **b** Immunohistochemistry images show the distribution of mCherry[+] ISNs and Dextran 488[+] retrogradely labeled axon-regrown spinal interneurons in 1-cut SCI fish with/without MTZ treatment. Injury sites are indicated by white arrows. **c–g** Ablation of ISNs by MTZ treatment prevents axon regrowth and locomotion recovery after SCI. Quantification of mCherry[+] ISNs numbers in the region of 500 μm around the injury site (**c**); free-swimming speed (**d**); maximum swimming speed (**e**); retrogradely labeled axon-regrown spinal interneuron numbers (**f**) and regrown axon density (**g**) after SCI with/without MTZ treatment. **h** Immunohistochemistry images show the distribution of mCherry[+] ISNs and Dextran 488[+] retrogradely labeled axon-regrown spinal interneurons in two-cut SCI fish with/without MTZ treatment after the second injury. **i–m** Effect of ISNs ablation in 2-cut SCI model. Mean data of mCherry[+] ISNs numbers in a 500 μm region covering the secondary injury site (**i**); free-swimming speed (**j**); maximum speed (**k**); retrogradely labeled axon-regrown spinal interneuron numbers (**l**) and regrown axon density (**m**) after the second injury with/without MTZ treatment. All data are presented as mean ± SD. *$P < 0.05$, **$P < 0.01$, ***$P < 0.001$, ****$P < 0.0001$, significant difference. For detailed statistics, see Supplementary Table 1.

(Fig. 3k)[37,46]. A large number of transcription factors related to neurogenesis and neuron differentiation were expressed in an all or none fashion in the injury-induced ISNs and the ISNs in the distal segments (Fig. 3l). Interestingly, genes related to neurite growth, such as Prepronociceptin gene (*pnocb*)[47,48], were over-expressed in the injury-induced ISN subpopulation (Fig. 3m; Supplementary Fig. 2m), implying increased remodeling of these ISNs while the postsynaptic adhesion molecule Slitrk3 (*slitrk3b*) which selectively regulates inhibitory synapse development[49] was downregulated (Fig. 3m), implying a reduction in inhibitory input. Further evidence of the transcriptomic separation of these two ISN subpopulations is seen in the expression of genes related to neurotransmitter transporters and ion channels (Fig. 3n, o). The hyperactivity displayed by terminal varicosities of the injury-induced ISNs likely reflects the increased expression of *scn1laa*, a voltage-gated sodium channel gene and *cacna1ia*, a T-type calcium channel gene compared to those ISNs located distal to the injury site (Fig. 3o). We also performed in situ in hybridization in *Tg(tph2: GFP)* fish at 4 weeks post injury. Four genes (*fezf2*, *her6*, *neurog1*, *pnocb*) were confirmed to be specifically expressed in the injury-induced ISNs, but not in ISNs in distal segments (Fig. 3p; Supplementary Fig. 2m). As a functional validation of the RNA-sequencing result, we used two-photon microscopy to monitor $Ca^{2+}$ dynamics in the presence of miberfradil (3 μM), a T-type $Ca^{2+}$ channel blocker, at the varicosities of injury-induced ISNs (Fig. 3e). The $Ca^{2+}$ oscillations at the varicosities of injury-induced ISNs were largely blocked by miberfradil (Fig. 3q) suggesting that upregulation of T-type calcium channel gene underlies the hyperactivity of ISNs.

In summary, the injury-induced ISNs form a functionally distinct subpopulation with a unique genetic profile that release serotonin constantly which in turn facilitates the axon regrowth of spinal interneurons and reestablishment of locomotor neural circuits.

**Serotonin facilitated axon regrowth of spinal glutamatergic interneurons via the 5-HT$_{1B}$ subtype.** To determine the serotonergic receptor subtypes mediating axon regrowth of spinal interneurons, we collected spinal cord tissue from five segments rostral and five segments caudal to the injury site in both uninjured and SCI fish at 1, 2, 4, and 6 weeks post injury for RNA-seq (Fig. 4a). Serotonergic system genes (transcription factors involved in serotonin neuronal differentiation, serotonin synthesis/transporter/receptor) were differentially expressed during recovery from SCI (Fig. 4b). Expression of *htr1b*, *htr2b*, and *htr7c* was depressed at 1 week post injury, then increased and returned to control levels at 6 weeks post injury (Fig. 4b; Supplementary Fig. 3a). In order to clarify the roles of these three receptor genes in axon regrowth of spinal interneurons, we generated *htr1b*, *htr2b*, and *htr7c* mutant animals using CRISPR/Cas9 technology. The target sites were located in exon 1. The mutation was predicated to cause a frameshift indel and produced a truncated

short peptide with a damaged seven-transmembrane G protein-coupled receptor domain (Supplementary Fig. 3b–d). These mutant animals are viable and adults swim normally. The mRNA levels of *htr1b*, *htr2b*, and *htr7c* were significantly reduced in mutant animals, suggesting the nonsense-mediated decay of mutant mRNAs (Supplementary Fig. 3b–d). In the recovery period after SCI, *htr1b*$^{-/-}$ animals had impaired locomotor function, poor rostral-caudal body coordination, markedly reduced numbers of axon-regrown spinal interneurons and decreased density of regrown axons at the injury sites (Fig. 4c, d; Supplementary Video). The reduced number of retrogradely labeled spinal interneurons was not due to decreased injury-induced neuronal regeneration after *htr1b* knockout (Supplementary Fig. 3e), revealing a specific effect of *htr1b* receptor on axon regrowth. In the *htr2b*$^{-/-}$ and *htr7c*$^{-/-}$ animals, axon regrowth and restoration of locomotor function after SCI were similar to that seen in wild-type animals (Supplementary Fig 3f). In a separate series of experiments, daily treatment of SCI *Tg(tph2:GFP)* fish from 3 days until 4 weeks post injury with 0.1 μM CP 93129 dihydrochloride, a selective 5-HT$_{1B}$ agonist, significantly accelerated the recovery process without altering the ISNs number (Supplementary Fig. 3g).

We characterized the proportion of axon-regrown excitatory and inhibitory interneurons in the second, third, and fourth segments rostral to the injury site using retrograde tracers in both *Tg(vglut2a:Gal4;UAS:GFP)* and *Tg(glyt2:GFP)* SCI animals. More than 80% of the axon-regrown neurons were *vglut2a*:GFP[+] excitatory interneurons, while *glyt2*:GFP[+] inhibitory interneurons accounted for less than 10% of the total (Fig. 4e, f). The ratio of excitatory to inhibitory interneurons was ten after SCI which was markedly greater than the ratio of two measured in uninjured animals (Fig. 4f) indicating a reorganization of spinal CPG neural circuit after SCI with more pronounced axon regrowth of excitatory interneurons than inhibitory interneurons. In order to determine whether 5-HT$_{1B}$ receptor acted as an injury-induced intrinsic factor preferentially promoting the axon regrowth of *vglut2a*:GFP[+] spinal excitatory interneurons, we analyzed the expression of *htr1b* in the axon-regrown spinal *vglut2a*:GFP[+] excitatory interneurons by FAC-sorting in SCI animals of *Tg(vglut2a:Gal4;UAS:GFP)* line at 4 and 8 weeks post injury (Fig. 4g). The expression level of *htr1b* in axon-regrown spinal *vglut2a*:GFP[+] interneurons (GFP[+]/RD[+]) increased significantly during this period and was much higher compared to that in non-regrown (GFP[+]/RD[−]) samples (Fig. 4h). This differential expression pattern was not observed in the uninjured animals (Fig. 4h Ctrl data), strongly suggesting that SCI induced upregulation of *htr1b* in axon-regrown spinal excitatory interneurons. Thus, *htr1b* is likely to be the 5-HT receptor subtype that facilitates axon regrowth of spinal interneurons and reestablishment of the locomotor circuit following SCI.

Finally, we explored how *htr1b* regulate axon regrowth of spinal *vglut2a*:GFP[+] interneurons using whole-cell patch-clamp

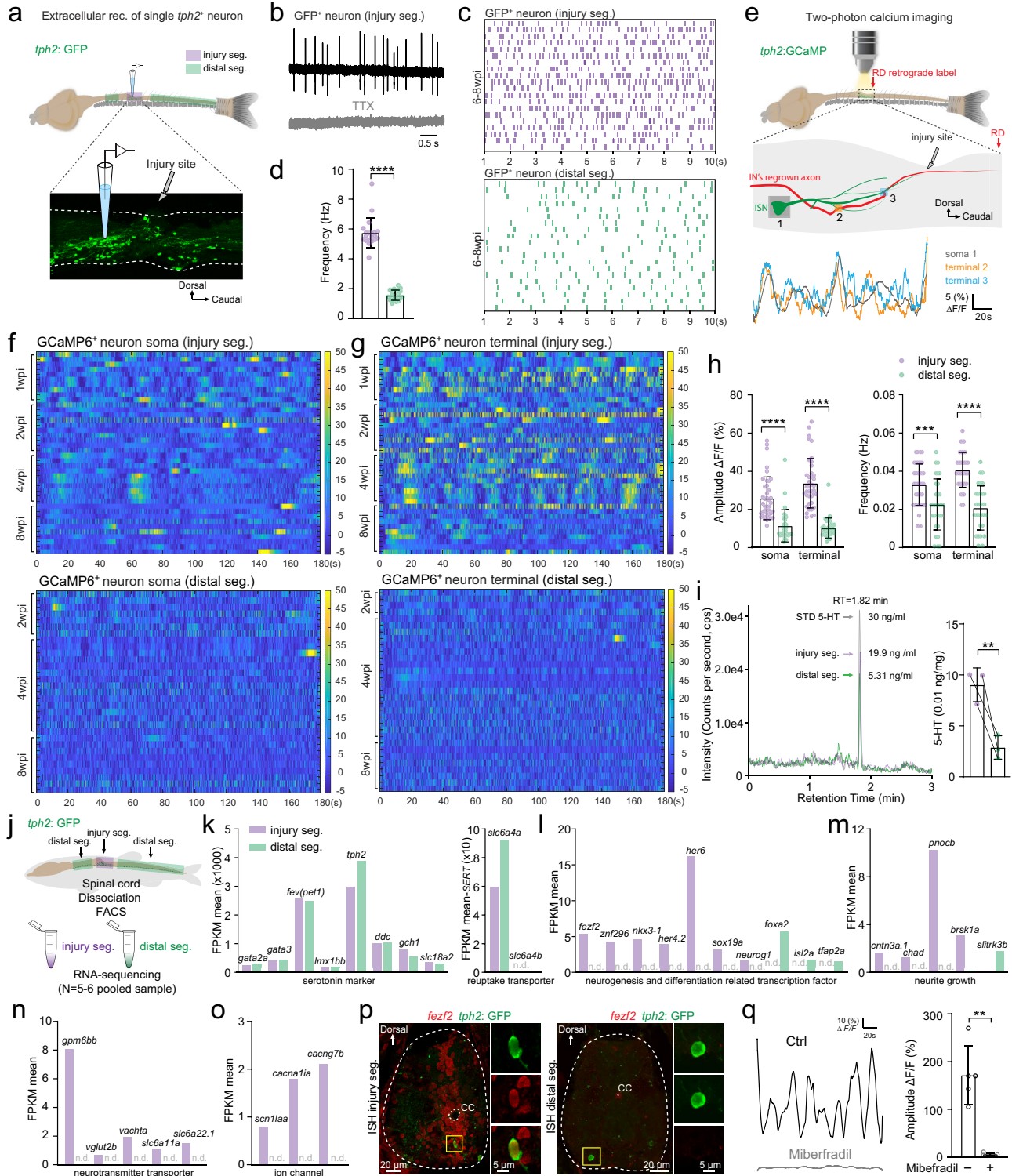

recordings and characterization of the intrinsic properties and morphologies of the axon-regrown *vglut2a*:GFP⁺ interneurons retrogradely labeled by RD in SCI fish of both *Tg(vglut2a:Gal4;UAS:GFP)* and *htr1b⁻/⁻ Tg(vglut2a:Gal4;UAS:GFP)* lines at 5 weeks post injury (Fig. 4i). Axon-regrown spinal *vglut2a*:GFP⁺ interneurons in *htr1b⁻/⁻* animals had significantly lower firing thresholds and slightly increased input resistance compared to those in *Tg(vglut2a:Gal4;UAS:GFP)* (Fig. 4j). The recorded axon-regrown *vglut2a*:GFP⁺ interneurons were filled with neurobiotin for morphological analysis. Remarkably, the axon thickness of

these *vglut2a*:GFP⁺ spinal interneurons in *htr1b⁻/⁻* animals was significantly thinner than that in the wild-type (Fig. 4k, l). These results indicate that *htr1b* is a critical regrowth-permitting molecule that is essential for serotonin-mediated axon regrowth and reconnection of disconnected spinal CPGs.

## Discussion

Descending brain motor commands initiate locomotor behaviors by activating spinal CPG[16,50]. Specific neuronal types and their

**Fig. 3 Injury-induced ISNs form a functionally distinct subpopulation actively releasing serotonin that acts on regrowing axons. a** Illustration of the in vitro spinal cord preparation for extracellular recording of the single GFP$^+$ ISN in the injury segment (purple) and distal segments (green). **b** Representative extracellular recording trace of a single injury-induced GFP$^+$ ISN spontaneous firing in the absence (upper) or presence (lower) of 1 μM TTX. **c, d** Raster plots (**c**) and quantification (**d**) of the spontaneous firing frequency of GFP$^+$ ISNs in the injury segment and distal segments within 10 s. **e** Two-photon calcium imaging is used to examine the Ca$^{2+}$ homeostasis (top) in in vitro spinal cord preparation. Typical examples of Ca$^{2+}$ changes (bottom) are recorded in the GCaMP6$^+$ ISN soma (gray) and two terminal varicosities (orange and blue) around the interneuron's regrown axon at the injury site (middle). **f, g** Heatmaps showing Ca$^{2+}$ oscillations during 3 min of GCaMP6$^+$ ISNs soma in the injury segment (**f** upper) and distal segments (**f** lower), and GCaMP6$^+$ ISNs terminal in the injury segment (**g** upper) and distal segments (**g** lower). **h** Quantification of Ca$^{2+}$ oscillations amplitude and frequency of GCaMP6$^+$ ISNs soma and terminal in the injury segment and distal segments. **i** (left) Representative LC–MS chromatograms showing retention time of standard serotonin (gray), extracellular fluid samples from injury segments (purple) and distal segments (green). (right) Quantification of serotonin released extracellularly per unit spinal cord tissue. **j** Sampling of GFP$^+$ ISNs from the injury segment and distal segments of the same SCI animal for FAC-sorting and bulk RNA-seq. **k–o** Mean FPKM values of common serotonin markers and reuptake transporter genes (**k**); differentially expressed (*P* value < 0.05) neurogenesis and neuron differentiation related transcription factors (**l**); neurite growth related genes (**m**); neurotransmitter transporter related genes (**n**); ion channel genes including calcium channel gene *cacna1ia* (voltage-dependent t-type calcium channel) (**o**) in GFP$^+$ ISNs from the injury segment and distal segments. n.d. denotes not detected. **p** In situ hybridization (ISH) for *fezf2* combining staining for GFP on cross sections of the injury and distal spinal segment, representing results from three independent experiments. Expanded images indicated by yellow boxes showing co-labeled cells. **q** Representative traces and mean amplitude of the robust Ca$^{2+}$ activity in GCaMP6$^+$ ISNs terminal in the injury segment in the absence (black) and presence (gray) of Calcium channel blocker miberfradil (3 μM). All data are presented as mean ± SD. *$P < 0.05$, **$P < 0.01$, ***$P < 0.001$, ****$P < 0.0001$, significant difference. For detailed statistics, see Supplementary Table 1.

synaptic connections define the locomotor characteristics[2,5,7]. Severe SCI damages the long-projecting axons of spinal interneurons located rostral to the injury site as well as neural circuits of local injured spinal segments and splits the spinal neural circuits into two. In this study, we took advantage of the regeneration of spinal cord after SCI in zebrafish and uncovered an intraspinal serotonergic mechanism that restores locomotion by promoting repair of the damaged spinal CPGs. We first identified a distinct subpopulation of injury-induced ISNs expressing unique transcription factors and discharging high-frequency action potentials, whose appearance was associated with the recovery process after SCI. Hyperactive terminal varicosities in these ISNs release serotonin constantly that acts on the 5-HT$_{1B}$ receptor to promote axon growth of long-projecting glutamatergic interneurons in spinal CPGs. We suggest that this is an intrinsic repair mechanism that facilitates the reconnection of disconnected spinal CPGs and promotes restoration of locomotor function after SCI. For decades, efforts have focused on revealing the cellular and molecular mechanisms by which the long descending axons regrow over the injury site but still the intrinsic mechanisms underlying reorganization of spinal neural circuits and gradual recovery of locomotor function after SCI remains unclear. Both brain-derived and propriospinal cocktail treatments have been shown to facilitate regrowth of all long-projecting axons and enhance the growth-supportive environment[51–57]. It has long been known that glutamatergic excitatory interneurons in spinal CPG are a key neuronal component in the generation of locomotion. Strong evidence suggests that recruitment of specific glutamatergic neurons is essential and indispensable for generation of required locomotion patterns[58,59]. One of the important findings in the present study is that after SCI over 80% of the axon-regrown spinal interneurons are glutamatergic, while glycinergic interneurons comprise less than 10% of the whole population. The ratio of long projecting glutamatergic interneurons to glycinergic interneurons greatly increased after SCI indicating that the reestablishment of spinal CPG neural circuits in SCI zebrafish mainly depends upon restoration of excitation mediated by axon regrowth of excitatory glutamatergic interneurons. The regrown long-descending axon of these rostral spinal excitatory interneurons project across both the injury segment as well as its caudal spinal segments to possibly synapse on the local spinal neurons, which reestablishes the locomotor neural circuit and recovers the pattern as well as rostral-caudal

coordination of locomotor activities. These spinal excitatory interneurons can drive the local excitatory interneurons and motor neurons via their regrown axon in caudal spinal segments to coordinate rostral-caudal activity. The left-right pattern could be achieved via activation of local spinal commissural inhibitory interneurons[58,59], which inhibit the neurons at the other side of spinal cord and enable the left-right locomotor pattern.

SCI caused upregulated expression of *htr1b* that was associated with axon regrowth and restoration of locomotion. Knock-out of *htr1b* impaired this recovery, and 5-HT$_{1B}$ agonist treatment facilitated it. Previous investigations have shown that the 5-HT$_{1A}$ receptor subtype negatively regulates axon regrowth of brain-derived long-projecting neurons after SCI in lampreys[60] by decreasing cyclic adenosine monophosphate (cAMP) levels[23], suggesting that dramatically different effects of serotonin on spinal interneurons and brain-derived neurons can arise through differential expression of 5-HT receptor subtypes. In addition to the upregulation of *htr1b* in spinal glutamatergic interneurons, a lack of descending serotonergic innervation after SCI is known to be a major limiting factor in the recovery of locomotor function in mammal[31,32]. In the current study, we show that intraspinal serotonin was supplied by an injury-induced ISN subpopulation, whose soma congregated around and along the injured spinal segments, and axons enwrapped and intertwined with the regrowing axons of spinal interneurons. We confirmed that these injury-induced ISNs are newly generated by co-labeling with EdU which labels proliferating cells. An earlier study in adult zebrafish showed that some of these regenerated ISNs are in close proximity to ependymo-radial glial cells, the spinal progenitor cells at the central canal and that regeneration of these ISNs appears to be promoted by a ventral midline-derived hedgehog signal[34]. Compared to those in distal spinal segments, the injury-induced ISNs comprise a distinct subpopulation expressing a number of unique transcription factors and genes relating to neuron differentiation, (e.g., *fezf2*, essential for the development of serotonergic neurons in the vertebrate brain[61], *gpm6bb*, a negative regulator of SERT trafficking[62]). Functional characterization by electrophysiology, Ca$^{2+}$ imaging and LC-MS analysis showed that the injury-induced ISNs discharge high-frequency action potentials and release serotonin constantly that promotes regrowth of axons of spinal glutamatergic interneurons.

In the mammalian CNS, serotonergic neurons in the brainstem project their axons over the brain and spinal cord[63]. However,

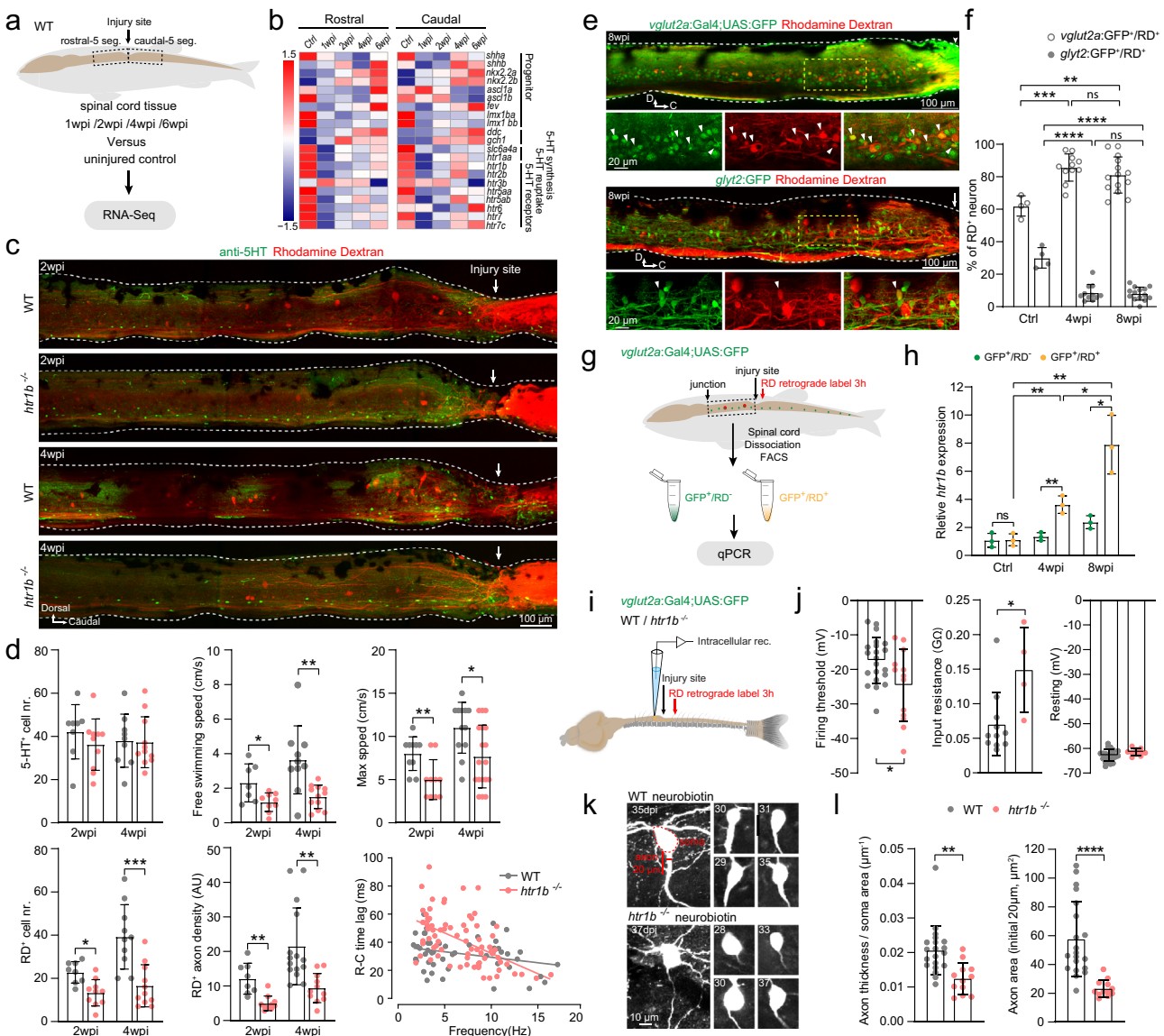

**Fig. 4 Serotonin facilitates axon regrowth of spinal interneurons via 5-HT₁B receptor. a** Collection of five spinal segments rostral or caudal to the injury site from wild-type (WT) SCI fish and corresponding segments from uninjured fish (n = 3 fish for each group) for RNA-seq (N = 4–5 libraries in each group). **b** Heatmap of mean FPKM values showing differentially expressed serotonin related genes after SCI. Differential expression analysis was performed between two groups (Ctrl versus 1-/2-/4-/6 wpi) using the DESeq2 (adjusted P-value < 0.05 after Benjamini and Hochberg's approach). **c** Immunohistochemistry images show the distribution of 5-HT⁺ ISNs and RD⁺ retrogradely labeled axon-regrown spinal interneurons in WT and htr1b⁻/⁻ SCI fish. **d** Quantification of WT (gray dots) and htr1b⁻/⁻ (red dots) SCI fish ISNs numbers in the region of 500 μm covering the injury site; free swimming speed; maximum speed; axon-regrown spinal interneuron numbers; regrown axon density and EMG recording of R–C time lag versus swimming frequency. **e** Immunohistochemistry images show the RD⁺ retrogradely labeled axon-regrown spinal interneurons in Tg(vglut2a:Gal4;UAS:GFP) (upper) and Tg(glyt2:GFP) (lower) SCI fish. Expanded images of yellow dash-line boxes show RD co-labeling with vlgut2a:GFP or glyt2:GFP (arrowheads) indicating axon-regrown excitatory or inhibitory interneurons. **f** Number of axon-regrown excitatory and inhibitory interneurons expressed as a percentage of RD⁺ neuron numbers after SCI, and comparison with the proportions of uninjured fish. **g, h** FAC-sorting of axon-regrown excitatory interneurons (vglut2a:GFP⁺/RD⁺) and non-labeled excitatory interneurons (vglut2a:GFP⁺/RD⁻) from uninjured and SCI fish (**g**) for detection of relative htr1b expression (**h**). **i** Illustration of the in vitro spinal cord preparation for whole-cell patch-clamp recording. **j** Mean data of electrophysiological properties of axon-regrown excitatory interneurons in SCI fish with WT and htr1b⁻/⁻ background. **k** Morphology of recorded axon-regrown excitatory interneurons filled with neurobiotin. Soma size (red dashed lines), the initial 20 μm length and thickness of axon (red line) are highlighted in top left image and used for morphology analysis. **l** Quantification of axon size of recorded axon-regrown excitatory interneurons. All data are presented as mean ± SD. *P < 0.05, **P < 0.01, ***P < 0.001, ****P < 0.0001, significant difference. For detailed statistics, see Supplementary Table 1.

these are not the only source of serotonergic innervation. In early studies, serotonergic ISNs were also revealed to exist in mammalian spinal cord. For example, they were found at sacral spinal cord in mouse[64], at thoracolumbar, sacral, and coccygeal spinal cord in adult rat[65] and frequently in the cervical spinal cord of monkey[66]. The ISNs displaying bipolar or multipolar morphology

were present in healthy as well as injured spinal cords in rats[67]. In mouse and monkey spinal cords, the ISNs were reported in an area ventral to the central canal[64,66] and similar groups of ISNs in the spinal cord of lower vertebrate have been suggested to modulate the function of locomotor CPG[25,68]. Further investigations should be performed to study the functional role of these

ISNs. Our findings suggest that local augmentation of intraspinal serotonergic system at specific levels within the injured spinal cord might provide the potential to reestablish spinal locomotor circuits in mammals. In vitro studies suggest that 5-HT$_{1B}$ receptor agonist administration promotes neurite extension and branching of cultured neurons[69,70]. These findings are limited by the whole body effect of administered pharmacological agents and also the lack of complexity of neurons in culture[71]. In the present study, we took advantage of the zebrafish SCI model with its remarkable regeneration ability to investigate in vivo the effect of the 5-HT$_{1B}$ receptor on axon regrowth of specific spinal interneuron. Using knockout animals, we demonstrated that the absence of 5-HT$_{1B}$ receptors impaired the axon regrowth of glutamatergic excitatory interneurons and recovery of locomotor function after zebrafish SCI.

In summary, this study highlights the importance of 5-HT$_{1B}$ receptors and suggest that injury induced ISNs are a potential cell therapy to facilitate reorganization of spinal CPG and restoration of locomotor function after SCI.

## Methods

**Animals**. All zebrafish (*Danio rerio*) were maintained in a zebrafish facility of Tongji University under standard conditions. Juvenile/adult zebrafish of both gender aged between 6 to 8 weeks and 1.5–2 cm in standard length[72] were used in the experiments. Wild-type and transgenic lines used in this study included *Tg(tph2:GFP)*[40], *Tg(tph2:Gal4)*[73], *Tg(UAS:GCaMP6)*[74], *Tg(UAS:nfsB-mCherry)*[75], *Tg(UAS:mCherry)*, *Tg(vglut2a:Gal4;UAS:GFP)*[76], *Tg(glyt2:GFP)*[10]. All experimental protocols were approved by the Animal Use Committee of Tongji University.

**Spinal cord transection**. Fish were anesthetized in 0.03% tricaine methane sulfonate (MS-222, Sigma-Aldrich). A complete transection of the spinal cord at the seventh segment from the brainstem and spinal cord junction was performed with a fine microsurgical stab knife (Surgical specialties) using a stereomicroscope. Successful transection was confirmed 24 h post injury by observation of the typical uncoordinated body movement.

To verify the role of local ISNs eliminating the descending serotonergic influence, a two-cut injury modal was employed, in which the spinal cord was completely transected a second time, this time at the fifth segment caudal to the first injury site 1 week after the primary injury.

**Neuron labeling and immunohistochemistry**. Labeling of regenerated neurons was done by intraperitoneal injection of 6 µL (10 mM) 5-ethynyl-2′-deoxyuridine (EdU, Sigma-Aldrich) daily after anesthesia with 0.03% MS-222 from day 1 to day 7 post injury. Retrograde labeling of spinal interneurons was performed using the fluorescent tracer Dextran Alexa Fluor 488 (10,000 MW, Invitrogen), RD (3000 MW, Invitrogen), or neurobiotin tracer (Vectorlabs). Fish at 2 to 8 weeks post injury were anaesthetized using MS-222 and the spinal cords were transected two segments caudal to the lesion site using fine scissors. The fluorescent dye or neurobiotin tracer was applied to the spinal cut using dye-soaked pins, and the animals were kept alive for 3–4 h to allow the retrograde transport of the tracer. Animals were anaesthetized in ice-water before fixation. After 24-h fixation with 4% paraformaldehyde (PFA), spinal cords were dissected out and washed three times each for 5 min with 1× PBS and permeabilized in 1× PBS containing 1% Triton X-100 for 2 h. After blocking with 5% BSA in 1% Triton in 1× PBS at RT for 1 h, spinal cords were then incubated with anti-GFP (Abcam ab1218, 1:2000 or ab290, 1:1000), anti-serotonin (Sigma-Aldrich S5545, 1:3000), anti-NeuN (Proteintech 26975-1-AP, 1:200), anti-Glutamate (Sigma-Aldrich G6642, 1:2000), anti-GABA (Sigma-Aldrich A2052, 1:1000), anti-ChAT (Millipore AB144P, 1:100, and streptavidin (Alexa Fluor 405/555 conjugate, Invitrogen) for 48 h at 4 °C. After rinsing with 1× PBS, the spinal cords were incubated with secondary antibodies at 4 °C overnight. Secondary antibodies: Alexa Fluor 647 Donkey anti-Mouse (Invitrogen A-31571 1:500), Alexa Fluor 488 Donkey anti-Mouse (Jackson Labs 715-545-150, 1:200), Alexa Fluor 488 Donkey anti-Rabbit (Jackson Labs 711-545-152, 1:200), DyLight 405 Goat anti-Rabbit (Jackson Labs 111-475-003, 1:200). After washing with PBS, the whole spinal cords were placed lateral side up on a slide and mounted with the anti-fading fluorescent mounting medium (Vectorlabs) for imaging. The BeyoClick EdU Cell Proliferation Kit (Beyotime Biotechnology) combined with Streptavidin-Alexa Fluor 555 (Invitrogen) was used to detect EdU signal according to the manufacturer's instruction. For NeuN staining, tissues were pretreated with Citrate Antigen Retrieval Solution (Sangon Biotech) for 30 min at 90 °C before treating with the primary antibody. Image acquisition was with an Olympus FV3000 laser scanning confocal microscope. The overview immunohistochemistry images are presented as maximum intensity z-projections of whole-mount spinal cord tissue. Images for detection of co-labeling (Fig. 3p,

Supplementary Fig. 1i, 2h, i, m) or co-localization (Supplementary Fig. 2l) represent a single plane of the region of interest.

**In situ hybridization**. In situ hybridization (ISH) was performed on 15 µm-thickness cross cryosections. Juvenile/adult *Tg(tph2: GFP)* fish were anesthetized using MS-222 and then eviscerated and fixed overnight in 4% FPA at 4 °C. Then PFA-fixed samples were equilibrated in 10, 20, and 30% sucrose in PBS overnight at 4 °C. On the following day, tissues were embedded in OCT compound (SAKURA), frozen at −80 °C, and sectioned serially at 15 µm using a Leica CM1860 UV cryostat. ISH on cryosections with digoxigenin (DIG)-labeled antisense RNA probes was carried out[77]. Probes used were *fezf2*[78], *her6*[79], *neurog1*[79], and *pnocb*[80]. Color reaction was performed using SIGMAFAST Fast Red (Sigma-Aldrich) after incubation with ant-DIG antibody conjugated to alkaline phosphatase (Sigma-Aldrich 11093274910, 1:3000). The sections were then washed in PBS containing 0.1% Tween20 and subjected to immunocytochemistry procedure for co-staining of anti-GFP.

**Measurement of locomotion behavior capacity**. Swimming behavior of fish at different time points after lesion was analyzed by video tracking of free-swimming fish. In each trial, a single fish was placed in a 9-cm open field dish and free-swimming traces were recorded using a digital camera (30 fps, EoSens CL, MIK-ROTRON) for 5 min. Video acquisition was done with DVR Express Core (IO Industries) and controlled by CoreView v2.1 software.

Maximum speed of injured fish was tested using a custom built forced swimming device(see Fig. 1d upper) similar to that used in an earlier study[81]. Zebrafish were put in groups in a swim tunnel equipped with an underwater flowmeter. Water current velocity was controlled using a pond pump. The fish were forced to swim against an initial flow of 3 cm/s and stepwise increments of 2 cm/s were made at 3-min intervals. The maximum speed of each fish was determined as the flow at which they were exhausted. Exhaustion was defined as when they were unable to swim away from the wire mesh at the end of the tunnel.

**Genetic ablation**. For ablation of serotonergic neurons, *Tg(tph2:Gal4)* fish were crossed with *Tg(UAS:nfsB-mCherry)*. After injury, the *Tg(tph2:Gal4;UAS:nfsB-mCherry)* double transgenic fish were treated with 5 mM metronidazole (MTZ, Sigma-Aldrich) according to the protocol in Fig. 2a and sampled at 2 and 4 weeks post injury. During the MTZ treatment, zebrafish were maintained in the darkness to avoid photo degradation. To preclude a non-specific effect of MTZ treatment, *Tg(tph2:Gal4;UAS:mCherry)* fish were used as control.

**Drug treatment**. After injury, the *Tg(tph2:GFP)* fish were treated with the potent and highly selective 5-HT$_{1B}$ agonist, CP 93129 dihydrochloride (Tocris)[82]. The drug was stored as a stock solution (100 mM) in DMSO (Sigma-Aldrich). It was added in the fish water tank 12 h daily at a final concentration of 0.1 µM from three days post injury until 4 weeks post injury according to the protocol in Supplementary Fig. 3g. Fish treated with the same concentration of DMSO were used as control.

**Electrophysiology and two-photon calcium imaging**. Injured zebrafish were anesthetized in a slush of frozen extracellular solution (134 mM NaCl, 2.9 mM KCl, 2.1 mM CaCl$_2$, 1.2 mM MgCl$_2$, 10 mM HEPES and 10 mM glucose, pH 7.8, 290 mOsm) containing MS-222 and were eviscerated. The brain-spinal cord was dissected out and transferred to the recording chamber and continuously perfused with extracellular solution. *Tph2*:GFP$^+$ serotonergic neurons (in the injury segment and distal segments) and retrogradely labeled RD positive spinal interneurons were visualized and targeted specifically using a fluorescence microscope (Scientifica) equipped with IR-differential interference contrast optics and a CCD camera with frame grabber. Whole-cell patch–clamp recording and extracellular single cell loose-patch–clamp recording were performed. For intracellular recordings, electrode filled with intracellular solution (120 mM κ-gluconate, 5 mM KCl, 10 mM HEPES, 4 mM Mg$_2$ATP, 0.3 mM Na$_4$GTP, 10 mM Na-phospho-creatine, pH 7.4 adjusted with KOH, 275 mOsm) was advanced into the spinal cord and approached the target neuron using the motorized micromanipulator while applying constant positive pressure. Intracellular signals were amplified with a MultiClamp 700B amplifier (Molecular Devices) and low pass filtered at 10 kHz. The recorded neurons were filled with neurobiotin for morphological analysis. For loss-patch recording, an electrode was filled with extracellular solution and moved forward with a small amount of positive pressure. After the electrode contacted the cell, the positive pressure was released and a gentle negative pressure was applied to make a loose patch onto the cell surface. Signals were recorded in current–clamp gap-free mode. In some experiments, TTX (1 µM, TOCRIS) was used.

For electromyogram (EMG) recordings during spontaneously swimming, injured zebrafish were anesthetized using MS-222 and the skin was removed to expose the muscle. The trunk of the fish was fixed on the recording chamber and continuously perfused with extracellular solution. One electrode was positioned on the intermyotomal junction rostral to the lesion site, the other two electrodes were placed opposing each other on the distal end caudal to the lesion site. Light suction

was applied, and signals were recorded using a Digidata series 1550B digitizer (Axon Instruments) and pClamp 10 software (Axon Instruments).

The brain-spinal cord of *Tg(tph2:Gal4;UAS:GCaMP6)* injured fish was prepared as described above. Intracellular $Ca^{2+}$ was monitored using a two-photon microscope (Scientifica) equipped with a 40× water-immersion objective (Olympus). GCaMP6 was excited using 900 nm light. Scan rate was 15.46 frames/s (1024 × 1024 pixels). Frame acquisition was performed using SciScan 1.2 software (Scientifica). A Z-projection stack was taken to show the position of target neuron in the spinal cord. In some experiments, the T-type calcium channel blocker miberfradil (3 µM, Sigma-Aldrich) was used. After recording, the preparation was fixed for immunostaining.

**LC-MS analysis**. Three spinal segments covering the injury site were collected and pooled from seven SCI zebrafish at 6 weeks post injury. Three caudal and three rostral segments distal to the injury site were collected and mixed from the same injured animal. Tissue from injury site and distal segments away from injury site were rinsed three times and transferred into a 0.6 ml microcentrifuge tube filled with 100 µL PBS for 30-min incubation at 28 °C. After centrifugation at 5000*g* for 5 min, the supernatant was collected for detection of released serotonin by LC–MS analysis. The residual tissue in the tube was dried and weighed for standardization of the serotonin concentration to sample weight. In total, 21 fish were used in three parallel groups. Standard and quality control samples were diluted 20-fold using blank PBS as used for tissue samples.

LC–MS analysis was carried out on a Triple Quad 6500+ LC–MS/MS System equipped with an ExionLC UHPLC unit (AB SCIEX, CA, USA). Chromatographic separation was achieved on a CORTECS HILIC Column (2.1x100mm, 2.7µm, Waters Corporation, MA, USA) at 35 °C. The mobile phase consisted of water (A) and acetonitrile (B), both containing 0.1% formic acid, at a flow rate of 0.6 ml/min. The 3-min elution gradient comprised the following: 0–0.5 min, 5% A; 0.5–1.1 min, 5–50% A; 1.1–2.17 min; 50% A; 2.17–2.2 min; 50–5% A and 2.2–3 min, 5% A. The injection volume was 0.5 µL. The mass spectrometric detection was performed using multiple reaction monitoring with an electrospray ionization source in positive mode. The ion spray voltage and temperature were set at 5500 V and 500 °C, respectively. The curtain gas was set at a flow rate of 35 psi; nebulizer gas and heater gas were both at 55 psi. CAD (collision gas) was set at 9. The MS conditions for detecting serotonin was Q1 Mass 177(m/z), Q3 Mass 160.1(m/z), entrance potential 10 V and collision cell exit potential (CXP) 10 V. Data acquisition and processing were performed with the Analyst software 1.7.1 from AB SCIEX.

**Fluorescence activated cell sorting**. Intraspinal serotonin neurons were isolated from the *Tg(tph2:GFP)* line. Spinal cord tissue from the injury area and distal segments were dissected out from the same fish, and digested using the Papain kit (Worthington Biochemical, 10U/ml in DMEM, osmolality of 285 mOsm) on a mechanical shaker at 37 °C, 300 rpm for 15 min and terminated by adding DMEM. Then the tissues were further dissociated by gentle pipetting up and down for 5 min. The resulting cell solution was filtered with a 40 µm cell strainer (BD Falcon) and Draq5 (Thermo Scientific) was added to check the viability. Subsequently, fluorescence activated cell sorting (FACS) was performed using a Beckman Coulter Moflo Astrios Cell Sorter (Beckman Coulter) and data were collected using Summit v6.3. The isolated *tph2*:GFP+ cells were collected and subjected to RNA-sequencing.

For purification of axon-regrown excitatory interneurons, the fluorescent tracer RD was retrogradely applied to *Tg(vglut2a:Gal4;UAS:GFP)* fish 4 or 8 weeks post injury. *Vglut2a*:GFP+/RD+ and *vglut2a*:GFP+/RD− cells were FAC-sorted from 40 spinal cords rostral to the lesion site at 4 or 8 weeks post injury and from 40 corresponding spinal cord segments collected from uninjured fish. Cells were placed in DMEM and were immediately subjected to RNA extraction using the ReliaPrep RNA Miniprep System (Promega) and reverse transcribed using the PrimeScript RT Reagent Kit (Takara). cDNA concentration was measured using the Qubit dsDNA high sensitivity assay kit (Invitrogen) and then sent for qRT-PCR analysis.

**RNA-sequencing**. Bulk RNA-Seq samples of ISNs from the injury area and distal segments respectively were FAC-sorted from *Tg(tph2:GFP)* line as described above. Total RNA was isolated using SMART-Seq™ v4 Ultra™ Low Input RNA Kit for Sequencing (Clontech). Sequencing libraries ($N = 5$–6) were generated using NEBNext Ultra™ RNA Library Prep Kit for Illumina following the manufacturer's instructions. All library preparations were sequenced on an Illumina Hiseq platform and 125 bp/150 bp paired-end reads were generated. Clean reads were obtained and mapped to the zebrafish genome using Hisat2. FPKM (Fragments Per Kilobase of transcript sequence per Millions base pairs sequenced) of each gene was calculated based on the length of the gene and reads count mapped to this gene. Differential expression analysis was performed using the DESeq R package. To increase the positive rate, genes with *P*-value (instead of adjusted *P*-value) < 0.05 were assigned as differentially expressed.

For RNA sequencing of spinal cord tissue, the five segments rostral and five segments caudal to the lesion site were collected from three fish at 1, 2, 4, and 6 weeks post injury. Corresponding segments were also collected from uninjured

animals. Total RNA was extracted using Trizol regent (Invitrogen), and cDNA libraries ($N = 4$–5) were subjected to Illumina sequencing according to the manufacturer's protocol. Differential expression analysis was performed using DESeq2. The resulting *P*-values were adjusted using the Benjamini and Hochberg's approach to decrease the false discovery rate. Genes with an adjusted *P*-value < 0.05 found by DESeq2 were designated as being differentially expressed.

**Generation of mutant zebrafish**. The mutant zebrafish lines were generated using CRISPR/Cas9 technology. The guide RNA target sites were designed using online tools (https://zlab.bio/guide-design-resources). Primers used for amplification of sgRNA templates were as follows: htr1b-sg-F: 5′-GTAATACGACTCACTATAGG TGCAACAGGTTATGTCTGGTTTTAGAGCTAGAAATAGC-3′; htr2b-sg-F: 5′-GTAATACGACTCACTATAGGCAAACAGACAGTGTGGATGTTTTAGAGCT AGAAATAGC-3′; htr7c-sg-F: 5′-GTAATACGACTCACTATAGGTGATTGGGA ATATGCTGGGTTTTAGAGCTAGAAATAGC-3′; sg-Common-R: 5′-AAAAGC ACCGACTCGGTGCC-3′. sgRNA was synthesized using the TranscriptAid T7 High Yield Transcription Kit (Thermo Scientific). Cas9 plasmid was digested with XbaI (Thermo Scientific) and capped Cas9 mRNA was synthesized using the mMESSAGE mMACHINE T7 Transcription Kit (Thermo Scientific). Totally, 300 pg Cas9 mRNA together with 30 pg sgRNA were injected into embryos at the one-cell stage. The primers used for genotyping were as follows: htr1b-Seq-F: 5′-TGCGTTTGTCATTGCCACCATTTC-3′ and htr1b-Seq-R: 5′-AGGGCGATGA GCAGCAGCG-3′; WT-htr1b-F: 5′-CTCAGACATAACCTGTTGCA-3′ and WT-htr1b-R: 5′-TTCAAAATGCGCTTCCTCGC-3′; MUT-htr1b-F: 5′-TGTCCTCAGC AGGTGGTCC-3′ and MUT-htr1b-R: 5′-TGAGTTGACGTAGCCAAGCC-3′; htr2b-Seq-F: 5′-ACGAAGAGAGTCTGTGACAATGC-3′ and htr2b-Seq-R: 5′-GG GTGTGTAATTGATGACAGAATTG-3′; htr7c-Seq-F: 5′-CCTCACACTTCTTC TACAACATCTC-3′ and htr7c-Seq-R: 5′-ACCTCTCCAAACAGCCACTTTCC-3′. The htr1b−/− mutant with *Tg(vglut2a:Gal4;UAS:GFP)* background was generated for analyzing the intrinsic properties of axon-regrown excitatory interneurons.

**Quantitative RT-PCR**. QRT-PCR was performed to detect the relative expression of *htr1b*, *htr2b* and *htr7c* using SYBR Green Realtime RCR Master Mix (TOYOBO) on LightCycler 96 system (Roche) by standard PCR following the manufacturer's instructions. Relative expression was normalized to *gapdh* levels. PCR primers were as below: *gapdh*, 5′-TTGTAAGCAATGCCTCCTG and 5′-CCCATCAACGGTCT TCTGT; *htr1b*, 5′-GCTTTCTACATCCCTACGC and 5′-GAAGTGAATCGGACA CAGTTA; *htr2b*, 5′-GGGTCTTTGGTGGCTTTC and 5′- AAAAAGGGAAGCCA GGCACT; *htr7c*, 5′-CGCCCTTTGACCTACCCT and 5′- GCCGAAGTCCTGGC TGAT.

**Data analysis and statistics**. Quantification of neuron and axon profiles (number, size, and density) were performed on confocal images using Fiji v1.53c (NIH). All the images were subjected to threshold adjustment such that the non-neuron background was reduced to zero leaving only fluorescent signals originating from cells. The numbers of axon-regrown interneurons, retrogradely labeled with tracer, were counted for half spinal thickness laterally within the second, third, and fourth segments rostral to the lesion. Tph2:GFP+ ISNs in the region of 500 µm surrounding the lesion site was calculated. The axon density was measured at the lesion site for half spinal thickness laterally using the YZ orthogonal view function of Fiji, corresponding to semi-cross section of the injury position (Supplementary Fig. 1e). The tracking of zebrafish free-swimming was carried out using the Python based free software ZebraZoom v1.17[83]. Electrophysiological data analysis was performed by Clampfit v10.6 (Molecular Devices) software. The R–C time lag, representing the activity propagation from injury rostral to caudal, was calculated using t2 (initial time of one swimming burst at L25) minus t1 (initial time of the swimming burst within the same cycle at L4) based on the drawing in Fig. 1e. The $Ca^{2+}$ response curves and heatmap were analyzed by a Matlab R2018a v9.4 (Mathworks) script. Photoshop (Adobe) and Fiji were used to process images. All data are presented as mean ± SD. One-way ANOVA or the student's unpaired or paired two-tailed *t*-test was used for statistical difference evaluation using Prism 8.0 (GraphPad Software), and *P* < 0.05 was considered to be statistically significant. For detailed statistics, see Supplementary Table 1.

**Reporting summary**. Further information on research design is available in the Nature Research Reporting Summary linked to this article.

## Data availability

All relevant data supporting this study are available from the corresponding authors upon reasonable request. Raw and processed RNA-seq data generated in this study are deposited into the GEO database with accession number GSE182911. Source data are provided with this paper.

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

## Acknowledgements

We thank Drs. T. Becker, J. Bruton, and X. Zhang for comments on the paper; Drs. A. El Manira, H.A. Burgess, J.-L. Du and China Zebrafish Resource Center (CZRC) for sharing the fish lines; Q. Zhang and R. Tang for fish line generation and care. This work was supported by the National Key Research and Development Program of China (2018YFA0108000), Shanghai Municipal Science and Technology Major Project (2018SHZDZX05), National Natural Science Foundation of China (31771168, 81811530025, 31972904, 31900706, and 8190126), Natural Science Foundation of Shanghai (20ZR1472500), the 2017 Thousand Youth Talents Plan of China (to J. Song), National Science and Technology Program Capacity Improvement Fund of Tongji University (190177, 4260141304/005), Shanghai Blue Cross Brain Hospital Co., Ltd., and Shanghai Tongji University Education Development Foundation.

## Author contributions

J.S. designed and initiate the project. C.H. did most of the experiments with help from N.N.G. J.M. helped generate the transgenic lines. Y.Z. performed extracellular recordings and in situ hybridization experiments. Z.W. and L.X. did the single-cell patch–clamp experiments. J.C. analyzed excitatory/inhibitory interneurons numbers. C.H. and N.N.G. analyzed the data. C.H., N.N.G., and J.S. wrote paper.

## Competing interests

The authors declare no competing interests.

## Additional information

**Peer review information** *Nature Communications* thanks Julien Bouvier and the other anonymous reviewer(s) for their contribution to the peer review this work. Peer reviewer reports are available.

