## [Peer Review File · Nature Communications]

Reviewers' Comments:

Reviewer #1:

Remarks to the Author:

In this manuscript by Huang, et al., the authors examine the identity and function of zebrafish intraspinal serotonergic neurons in recovery from spinal cord injury. Using molecular, genetic, and physiological approaches, they show that injury-induced ISNs have unique gene expression and electrophysiology compared to ISNs in uninjured spinal cord segments, and constitutively release serotonin on Htr1b receptors residing on neighboring excitatory interneurons. The authors further demonstrate that htr1b is required for normal axon regrowth and functional recovery of locomotor behavior after injury. They conclude that injury-induced ISNs and serotonin signaling comprise an important mechanism for SCI recovery.

The manuscript is generally clear and well-written, and the data are exciting and comprehensive. I believe this work will be of high interest to others in the field and also to a more general audience. In my opinion, the most novel and significant findings are the identification of htr1b as an essential component of axon regrowth and function in injured neurons, and the specific properties of 5-HT-producing neurons generated after injury. There are only a few experimental issues that I think should be addressed before publication, along with some minor suggestions for the manuscript itself.

- Based on MTZ ablation of tph2+ cells in their 2-cut model, the authors conclude that injury-induced ISNs can independently promote axon regrowth, etc. in the absence of descending 5-HT modulation (lines 191-5). However, it would seem that a necessary control would be to similarly ablate some other cell type that is not hypothesized to function in regeneration. This would rule out the possibility that the act of cell ablation itself, which likely induces an immune response and debris clearance, is not sufficient to impair regeneration and recovery.

-The characterization of "putative synapses" based on the intersection of 5-HT+ axon varicosities with regenerating glutamatergic axons seems somewhat arbitrary. Neuromodulators are often released by dense core vesicles at SV2-negative extrasynaptic sites, which do not include canonical postsynaptic proteins on the receiving axons. The presence of Htr receptors at these sites would certainly be a strong indication of serotonin signaling, but again this would not necessarily meet the rigorous definition of a synapse.

- The question of synaptic vs. extrasynaptic release sites of 5-HT raises a second issue of whether the ISNs co-release any other neurotransmitters. Do the authors have any data indicating co-release of glutamate, GABA, or anything else with serotonin?

- In lines 73-4, it is not clear what the authors mean by saying that "a subpopulation of serotonergic neurons is preferentially activated to initiate and promote the repair and recovery process." In the references they cite, are they referring to physiological activation, or some other change in the properties of existing 5-HT+ neurons?

- In line 114, the authors claim that the number of ISNs decreases over 1 year post-injury, "strongly implicating their important role in promoting spinal cord regeneration". It does not seem that this decrease alone makes a strong argument for a role in regeneration, as several other possible functions cannot be ruled out. For example, maybe the ISNs play a post-regenerative role in plasticity or tissue remodeling.

- Several minor grammatical and spelling mistakes occur throughout the manuscript, and can easily be fixed by a native English-speaker.

Reviewer #2:

Remarks to the Author:

This paper investigates mechanisms that can underlie the recovery of movements following spinal cord injury, a major public health issue with still no cure. Following an experimental injury in the

zebrafish model, the work identifies that the injury favors the generation of intra-spinal neurons (ISNs) releasing serotonin. These, in turn, may promote axonal regeneration of glutamatergic neurons that are possibly involved in generating swimming activity and these changes may support the regain of swimming capacity. Although a propriospinal source of serotonin was previously demonstrated following injury, the most important and novel findings reported here are i) the injury-induced 5-HT population has important differences in gene expression from the endogenous 5-HT neurons, ii) ablating such ISNs functionally limits locomotor recovery, and iii) the beneficial effect of ISNs is independent from supra-spinal sources serotonergic. The methodological approaches and analytic tools are extremely valid and adequate, and appear to be fully mastered. I think in fact that the complementarity of approaches is a major strength of this work. The 2-cut injury model is also a very elegant and a very meaningful strategy to isolate propriospinal versus descending serotonergic action. Altogether, this study is well performed, contains substantial novel data in the field, and clearly deserves to be considered further.

There are however multiple elements that must be strengthened. The first one is that the manuscript is globally poorly-written. Detailed comments on this below. There are also some conceptual/methodological caveats that require some additional work.

Major comments:

1. English MUST be SUBSTANTIALLY improved. Currently, the paper is far from a publication quality. Improper writing starts from first line of the abstract with many missing words (injuries "to the" spinal generator) and this continues throughout the manuscript. This often prevents a clear understanding of what the authors mean. Please have the revised paper thoroughly reviewed by a native speaker. Beyond grammar issues, the logical transitions between paragraphs should be better motivated better, scientifically. There are also multiple sentences that are very confusing and unclear, including in the methods. I cannot list them all here, but I insist that these are not subtle and isolated problems, but that there is an ABSOLUTE need for improvements.

2. Related to the above, there are also some issues with the use of the terms "regeneration" and "regrowth". Eg line 77: what is "motor neuron regeneration"? An increase in the number of motor neurons, or a regrowth of their severed axons? I think that the problem is that authors use the term regeneration to refer either to the augmented number of ISNs (meaning newly-born cells), but also to refer to axonal regrowth of severed axons. Authors must be careful when using these terms. Rather, I would advise to use dedicated terms for new cells versus regrown axons, and/or to define these terms more clearly. Currently, it's confusing whether "regenerated" means the regrowth of a compartment of a pre-existing neuron or a newly-generated neuron.

3. Not a single experiment here ascertains that increased serotonin concentration in the cord from hyperactive injury-induced ISNs is underlying improved swimming. Authors mention some previous work having established the importance of 5HT release, the lack of receptors does indeed impair recovery, and silencing of Tph1 neurons also impairs recovery, so it is of course a plausible explanation. Yet, the importance of the present work is to claim to have identified the source of 5-HT that supports the recovery: injury-induced ISNs. Therefore, I feel that it is essential to causally link the increased number of ISNs to an actual increase of 5-HT neurotransmission. Figure 3A-I touches upon this question, but there is no hallmark of serotonin concentration in the cord, or release by the neurons. Likewise, synaptic contacts are detected from Tph1-reporter axons (Figure 3I), but no demonstration that these contacts release serotonin is provided. Authors must document this matter experimentally. Currently, the claim that "ISN constantly release serotonin" (line 195, 230, 327, 365, 805 and elsewhere) is not valid. This is all the more important since the origin of these injury-induced ISNs neurons is not clear (see other comment on this). Authors mention they could arise from a neurotransmitter switch. This also leaves open the possibility that ISNs can co-release other transmitters. Hence, authors MUST also firmly exclude that ISNs co-release other neurotransmitters, such as glutamate which is another important activator of swimming CPGs.

4. A large part of the work relies on comparing the ISNs at the lesion site with other 5HT neurons

located more distally, the latter serving as controls. This is a rather peculiar strategy in my opinion and non-lesioned animals would have appeared a better option. Since animals do recover whole-body swimming, isn't it possible that distally-located 5HT neurons post-lesion are also influenced by the injury-induced ones or by the global changes in spinal circuits, and may thus be functionally distinct to the ones in non-lesioned fish? Can the authors provide comparisons with non-lesioned fish and clarify this?

5. Related to the above, I am confused with the strategy used to claim that ISNs are newly-generated neurons, which is an important finding of the work.

(a) Their number undoubtedly increases at the lesion and significant fraction of them is captured by the EdU injection (Figure S1D). But since the zebrafish at six to eight weeks are still growing (see also minor comment on age), there might be cells in proliferation even in non-lesioned animals. Even if 50% of Tph2::GFP cells incorporated EdU, this does not mean that there is more proliferation than in non-injured animals, and thus that this proliferation is linked to the injury. Authors must report tph2:GFP+/ EdU+ cells in animals without injury and/or at the distal segments where there is presumably no newly-generated ISNs.

(b) I didn't get the rationale for looking at EdU incorporation in the 1-cut and 2-cut model (Fig S1D). Where is the sampling made in the 2-cut model, at the first cut or second cut, or distally? This is not explained/discussed.

6. Related to the above, I would also appreciate some discussion lines regarding the mechanisms that allow ISNs to increase in number (possibly to proliferate) following the injury. Is there a latent source of progenitor cells in the cord, which is engaged into proliferation by lesional tissue? Or is it due a transformation, close to the lesion, of pre-existing cells that would change their neurotransmitter? The latter would make it all the more important to verify co-release of other transmitters.

7. I would also appreciate a little bit more background on ISNs. In mammals, there is somewhat of a consensus that the source of serotonin is mostly (exclusively?) supra spinal. Any evidence of intra-spinal serotonin release, either in health or following SCI in mammals too? Also what is known about the projection pattern (e.g., local, ascending, commissural) of ISNs? If one were to discuss the relevance of the findings in rodents, in which the locomotor circuits are more confined to specific segments, do authors think that increasing 5-HT locally at an injury site more rostral, would impact these lumbar circuits? Some discussions along these lines would help to expand the usefulness of the findings.

8. The differential impact of ISNs on excitatory and inhibitory neurons is very interesting. Yet, how this may impact locomotion is not clearly discussed. Both inhibitory and excitatory circuits are required for a proper left-right and inter-segmental coordination of swimming movements, so I'm not sure about the conclusion that "the reestablishment of spinal CPG neural circuits in adult SCI zebrafish is mainly dependent on restoration of excitation mediated by axon regrowth of excitatory glutamatergic interneurons" (line 345). This needs to be clarified and discussed further. Also, the control data (Figure S3G) should be incorporated into the main Figure 4F to allow comparison between the lesioned and non-lesioned animals. Also a typo in that figure: Uninjured (instead of uninjury).

9. I didn't find the supplemental video to be convincing at all. It is very short, both fish do a similar number of swimming bouts and the difference in distance travelled is not evident. Also, a scale bar is needed in that video. Please revise it. The data in the Figure 4D is convincing though.

Minor:

1. At multiple times in the manuscript, authors mention that some variables (number of ISNs, swimming behavior, receptor expression levels) at 8 wpi reach the number present before injury. When taking the example of the number of ISNs (line 113), I'm not sure this is what the statistics say in the figure (1B). I see that there is indeed a plateau and a then decrease but the numbers still is significantly higher than pre-injury. Same for the expression of 5-HT receptors (line 269, Figure 4B). Please be more transparent here; that these values stay modulated from the control state does not invalidate the present findings.

2. Please indicate or represent the sites of dextran injections on histology figures (eg, Figure 1G, K, and others) so that the reader can easily appreciate the directionality of the labelling. Also include the rostral-caudal and D/V axis on all lateral views of the spinal cord (few are missing).
3. Some questions about age and growth: it is claimed that young adult zebrafish are used, between six to eight weeks and 1.5 to 2 cm in length (line 384). (a) Why was this stage chosen? (b) The authors may want to avoid "adult" because their study model is between 2 stages: juvenile and adult. (c) The authors should specify if the length is the standard length (SL) or the total length (e.g Growth and maturation in the zebrafish, *Danio rerio*: a staging tool for teaching and research, Singleman and Holtzman, 2014, PubMedID : 24979389). (d) Since zebrafish have allometric growth, is it relevant to count tph2+ cells in the same region of 500µm surrounding the lesion site at all the stages?
4. Maybe say once from the beginning of the result section that this is a complete transection.
5. Showing the acute situation (cord separated) in Figure 1B would help. Likewise, it is claimed that "when the transected spinal tissues were still separated, no retrogradely labeled interneuron was observed" (line 139). This should be shown and quantified, to ascertain the completeness of the lesion.
6. line 247: there is no evidence here for a "functional" separation of the subtypes (transcriptomic data).
7. Figure S1C is missing the situation in control (unlesioned or pre-lesioned) animals. This is needed to claim that the numbers reach control values over time.
8. Figure 2B, 2H: typo at Dextran488.
9. Figure 2, 3 4, 3B, S1, S2D (possibly elsewhere), and associated text: it is incorrect to refer to tph+ cells. These are mCherry+/GFP+ neurons on a reporter fish line.
10. Transcriptomics data (Figure 3K-O): I think that including a broad category of marker genes on the figure panels (e.g. 5-HT transmission, neuronal differentiation,...) would help.
11. Line 140: It cannot be claimed that similar results were observed using another tracer, neurobiotin (Fig S1E) since neurobiotin cells were only counted at 2wpi (while dextran cells were counted at 2, 4 and 8 wpi). I am not sure the neurobiotin data altogether adds much.

Reviewer #3:

Remarks to the Author:

In this manuscript by Huang et al., the authors investigate the contribution of local serotonergic neurons in the regeneration of the spinal cord – by opposition to central descending neurons – and in locomotor behavior in zebrafish. While the topic covered by this study is overall of high interest, many conclusions are not supported by the data presented in the manuscript and appear overstated.

Major limitations impeding the publication of this manuscript in its current state are listed below.

The first part of this work is designed to provide evidence that spinal cord injury (SCI) induces the regeneration of local serotonergic neurons that are important for locomotor recovery (Figure 1 title "Injury induced regeneration of intraspinal serotonergic neurons (ISNs) is positively correlated with locomotor recovery"). However, the data presented in Figure 1 do not support this conclusion:

a. The optimization of a double spinal transection model is well suited to determine the generation of local serotonergic neurons and their role in motor recovery. However, no clue is provided on the recovery of locomotor patterns in this 2-cut model.

b. No clue is given as to whether the regenerated axons (labelled with rhodamine dextran) are indeed the axons of ISNs, which is the main conclusion provided for this figure (as the title is formulated). Given the results shown in Figure 1K, no colocalization between rhodamine and GFP positive tracts of tph2 neurons can be detected. This rather suggests that axon regeneration and accumulation of tph2 positive cells occurring locally between the two injury sites are two correlated events targeting different cell populations.

Second, data obtained upon chemogenetic ablation is poorly convincing. While one would expect a large impact of tph2 neurons ablations on locomotor capacity, the amplitude of the effects observed is minimal in the 2-cut injury model (Figure 2J, 2K). Moreover, no control regarding the potential effect of metronidazole on control animals is provided to fully validate the experimental design and demonstrate that metronidazole per se does not affect spinal cord regeneration. Indeed, metronidazole, an antibiotic with well-known anti-inflammatory properties, could act directly on axon regeneration upon SCI and hinder the contribution of local ISNs. This is further supported by earlier reports suggesting that metronidazole interferes with serotonin levels in the nervous systems (Karamanakos PN, Pappas P, Boumba VA, Thomas C, Malamas M, Vougiouklakis T et al. Pharmaceutical agents known to produce disulfiram-like reaction: effects on hepatic ethanol metabolism and brain monoamines. *Int. J. Toxicol.* 2007; Karamankos, The possibility of serotonin syndrome brought about by the use of metronidazole. *Minerva Anesthesiol.* 2008).

Third, the strategy implemented to identify the molecular signature of local ISNs at the injured spinal segments does not seem valid. Indeed, the authors compare the repertoires of transcripts of GFP+ tph2 neurons depending on their location respective to the injury site, instead of cross-comparing the transcripts enriched against dark cells in each region (i.e. repertoire of GFP+ versus dark in injured segments against the repertoire of GFP+ versus dark in distal segments). Moreover, none of these identified transcripts is validated (using in situ hybridization to compare expression patterns in injured and distal segments).

Fourth, the increased calcium variations in ISNs (in amplitude and frequency, Figure 3H) combined with a co-detection of tph2:GCaMP and SV2 is not sufficient to conclude that "ISNs [...] actively release serotonin that acts on re-growing axons" (title of Figure 3) as no measurement of presynaptic vesicle release probability is provided. Furthermore, there is no specification as to whether Figure 3I represents a single plane or a stack projection, which is critical for the interpretation of this anatomical argument stating about the apposition of serotonergic varicosities onto rhodamine labelled axons.

Finally, the generation of htr1b 2b and 7c mutants is designed to provide evidence on the contribution of local serotonergic signaling on axon regrowth and locomotor recovery. However, both of these phenotypes are not impacted in the two latter mutants (Figure S3E). Moreover, the locomotor phenotype of htr1b mutants is minimal (Figure 4D, the distribution of free swimming speeds and max speeds overlap very much) and even undetectable from Video S. Then, what could explain the absence of phenotype in htr2b and 7c mutants while they appear as the most regulated target transcripts after SCI (Figure 4B)? Could gene compensation in the generated mutants hinder phenotypic defects?

Minor comment:

In general, the manuscript would benefit from text and language editing and from a better description of the experimental paradigms / analysis methods to gain clarity.

The authors assume in the main text that the targets of ISNs are spinal interneurons. This is stated early in the text (line 167, 185, 193, 221), while evidence for this statement is provided only in Figure 4, which is very misleading.

Reviewer #1 (Remarks to the Author):

In this manuscript by Huang, et al., the authors examine the identity and function of zebrafish intraspinal serotonergic neurons in recovery from spinal cord injury. Using molecular, genetic, and physiological approaches, they show that injury-induced ISNs have unique gene expression and electrophysiology compared to ISNs in uninjured spinal cord segments, and constitutively release serotonin on Htr1b receptors residing on neighboring excitatory interneurons. The authors further demonstrate that htr1b is required for normal axon regrowth and functional recovery of locomotor behavior after injury. They conclude that injury-induced ISNs and serotonin signaling comprise an important mechanism for SCI recovery.

The manuscript is generally clear and well-written, and the data are exciting and comprehensive. I believe this work will be of high interest to others in the field and also to a more general audience. In my opinion, the most novel and significant findings are the identification of htr1b as an essential component of axon regrowth and function in injured neurons, and the specific properties of 5-HT-producing neurons generated after injury. There are only a few experimental issues that I think should be addressed before publication, along with some minor suggestions for the manuscript itself.

We thank the reviewer for the comments. We took the reviewer's suggests and meticulously revised our manuscript.

- Based on MTZ ablation of tph2+ cells in their 2-cut model, the authors conclude that injury-induced ISNs can independently promote axon regrowth, etc. in the absence of descending 5-HT modulation (lines 191-5). However, it would seem that a necessary control would be to similarly ablate some other cell type that is not hypothesized to function in regeneration. This would rule out the possibility that the act of cell ablation itself, which likely induces an immune response and debris clearance, is not sufficient to impair regeneration and recovery.

Response: Done. We provide several lines of evidence to address this question.

First, to rule out the influence of MTZ-induced cell ablation on the axonal regrowth and locomotor recovery, we performed control experiments in Tg(*olig1:Gal4;UAS:nfsB-mCherry*) transgenic line, where we ablated the *olig1:mCherry+* cells after MTZ treatment, since the *olig1* gene has not been reported to be involved in axonal regrowth and locomotor recovery after injury. The results indicate that ablation of *olig1:mCherry+* cells had no effect on the regrowth of axon and locomotor recovery. The number and axonal density of axon-regrown neurons were not affected by MTZ induced cell ablation. We have not put these data in the revised manuscript due to shortage of space. We illustrate these data in the Figure

below.

Figure A: Immunohistochemistry images of 1-cut whole-mount spinal cords showing the distribution of mCherry+ olig1 cells (red) and retrogradely labeled axon-regrown spinal interneurons (green) of *Tg(olig1:Gal4;UAS:nfsB-mCherry)* 4wpi SCI fish with/without MTZ treatment. **Figure B:** Quantification of 4wpi SCI *Tg(olig1:Gal4;UAS:nfsB-mCherry)* fish maximum swimming speed; retrogradely labeled axon-regrown interneuron numbers and regrown axon density with (red dots)/without (grey dots) MTZ treatment. Each dot represents one fish.

Second, we performed experiments to preclude any effect of MTZ itself on axonal regrowth and locomotor recovery and added these data in Supplementary Fig. 1k. In these experiments, MTZ treatment was applied to *Tg(tph2:Gal4;UAS:mCherry)* transgenic line instead of the *Tg(tph2:Gal4;UAS:nfsB-mCherry)* transgenic line, we did not perform ablation of any cell types. This shows that MTZ alone did not impair axon regrowth and locomotor recovery (Page 9 Line 185-189).

Finally, MTZ induced chemo-genetic ablation is widely used for regeneration research in the central nervous system of both larval and adult fish (Montgomery et al., 2010; Ohnmacht et al., 2016; Shimizu et al., 2015). Combined with the new results in our revised manuscript, we consider that the effect of MTZ induced chemo-genetic ablation in the *Tg(tph2:Gal4;UAS:nfsB-mCherry)* transgenic line on axonal regrowth and locomotor recovery is through specifically ablating serotonergic neurons.

-The characterization of “putative synapses” based on the intersection of 5-HT+ axon varicosities with regenerating glutamatergic axons seems somewhat arbitrary. Neuromodulators are often released by dense core vesicles at SV2-negative extrasynaptic sites, which do not include canonical postsynaptic proteins on the receiving axons. The presence of Htr receptors at these sites would certainly be a strong indication of serotonin signaling, but again this would not necessarily meet the rigorous definition of a synapse.

Response: Done.

Serotonin released both extrasynaptically and intrasynaptically from varicosities of

the regenerated ISNs can spread throughout the injury site and bind to Htr1b receptors on the regrown axon of spinal interneurons. We have rephrased this section and avoid the emphasis on the intrasynaptic release indicated by the SV2 protein labeling.

We examined the extracellular concentration of released serotonin via LC-MS method. The concentration of released serotonin in the injury segment was much higher than that measured in distal segments (Fig. 3i, Page 10 Line 223-227). The high concentration of extracellular serotonin may indicate that the serotonin was released mainly from extrasynaptic sites. The released serotonin, in high concentration, can bind to Htr1b receptors on the regrown axon at the injury sites.

In addition, we found that the densely distributed axon of the regenerated ISNs displayed close apposition to the regrown axon of spinal interneurons, which would favor diffusion of high concentration of released serotonin to the regrown axon.

- The question of synaptic vs. extrasynaptic release sites of 5-HT raises a second issue of whether the ISNs co-release any other neurotransmitters. Do the authors have any data indicating co-release of glutamate, GABA, or anything else with serotonin?

Response: New experiments were done to address this point. Anti-Glutamate, anti-GABA and anti-ChAT staining were done in 1-cut *Tg(tph2:GFP)* fish at four weeks post injury (Supplementary Fig. 2h-j). We quantified the numbers of double-stained serotonergic neurons in the injury segments. About 6% *tph2:GFP*⁺ neurons were positive for anti-Glutamate and 6% *tph2:GFP*⁺ neurons were positive for anti-GABA. The numbers of double-stained neuron revealed by anti-Glutamate and anti-GABA labeling were 3 to 4 per injury segment and almost undetectable for anti-ChAT labeling (Page 10-11 Line 227-231). Thus, the dominant neurotransmitter released by regenerated ISNs is serotonin, which can facilitate the axonal regrowth and locomotor recovery via activation of Htr1b receptors.

- In lines 73-4, it is not clear what the authors mean by saying that “a subpopulation of serotonergic neurons is preferentially activated to initiate and promote the repair and recovery process.” In the references they cite, are they referring to physiological activation, or some other change in the properties of existing 5-HT+ neurons?

Response: This was our mistake and we thank you for pointing it out. The reference we cited suggested that 5-HT⁺ neurons show molecular and cellular diversity under physiological condition. We intended to say that the regenerated ISNs might be a different subpopulation of serotonergic neurons from the existing ones, since they were newly regenerated right after spinal cord injury. The molecular and cellular properties would be distinct from the existing ones, which might help identify

regenerated ISNs as a distinct subpopulation.

To avoid confusion, we rephrased the sentence in the revised manuscript. Please see Page 3-4 Line 65-68.

- In line 114, the authors claim that the number of ISNs decreases over 1 year post-injury, “strongly implicating their important role in promoting spinal cord regeneration”. It does not seem that this decrease alone makes a strong argument for a role in regeneration, as several other possible functions cannot be ruled out. For example, maybe the ISNs play a post-regenerative role in plasticity or tissue remodeling.

Response: We entirely agree with the reviewer that the change in ISNs numbers at later stages of spinal cord injury would not strengthen their role in regeneration. It is possible that these ISNs still take part in other process associate with regeneration at the later stage during the recovery. To avoid confusion, we removed the speculation about the numbers of ISNs in the long term.

- Several minor grammatical and spelling mistakes occur throughout the manuscript, and can easily be fixed by a native English-speaker.

Response: We employed a native English-speaker to correct the grammatical mistakes in this manuscript.

Reviewer #2 (Remarks to the Author):

This paper investigates mechanisms that can underlie the recovery of movements following spinal cord injury, a major public health issue with still no cure. Following an experimental injury in the zebrafish model, the work identifies that the injury favors the generation of intra-spinal neurons (ISNs) releasing serotonin. These, in turn, may promote axonal regeneration of glutamatergic neurons that are possibly involved in generating swimming activity and these changes may support the regain of swimming capacity. Although a propriospinal source of serotonin was previously demonstrated following injury, the most important and novel findings reported here are i) the injury-induced 5-HT population has important differences in gene expression from the endogenous 5-HT neurons, ii) ablating such ISNs functionally limits locomotor recovery, and iii) the beneficial effect of ISNs is independent from supra-spinal sources serotonergic. The methodological approaches and analytic tools are extremely valid and adequate, and appear to be fully mastered. I think in fact that the complementarity of approaches is a major strength of this work. The 2-cut injury model is also a very elegant and a very meaningful strategy to isolate propriospinal versus descending serotonergic action. Altogether, this study is well performed, contains substantial novel data in the field, and clearly deserves to be considered further.

We thank the reviewer for the positive comments on our manuscript.

There are however multiple elements that must be strengthened. The first one is that the manuscript is globally poorly-written. Detailed comments on this below. There are also some conceptual/methodological caveats that require some additional work.

Major comments:

1. English MUST be SUBSTANTIALLY improved. Currently, the paper is far from a publication quality. Improper writing starts from first line of the abstract with many missing words (injuries “to the” spinal generator) and this continues throughout the manuscript. This often prevents a clear understanding of what the authors mean. Please have the revised paper thoroughly reviewed by a native speaker. Beyond grammar issues, the logical transitions between paragraphs should be better motivated better, scientifically. There are also multiple sentences that are very confusing and unclear, including in the methods. I cannot list them all here, but I insist that these are not subtle and isolated problems, but that there is an ABSOLUTE need for improvements.

Response: We appreciate the reviewer’s kind concerns. We employed a native English-speaker to check the grammar and spelling in the revised manuscript.

2. Related to the above, there are also some issues with the use of the terms “regeneration” and “regrowth”. Eg line 77: what is “motor neuron regeneration”? An increase in the number of motor neurons, or a regrowth of their severed axons? I think that the problem is that authors use the term regeneration to refer either to the augmented number of ISNs (meaning newly-born cells), but also to refer to axonal regrowth of severed axons. Authors must be careful when using these terms. Rather, I would advise to use dedicated terms for new cells versus regrown axons, and/or to define these terms more clearly. Currently, it’s confusing whether “regenerated” means the regrowth of a compartment of a pre-existing neuron or a newly-generated neuron.

Response: Thank you for pointing out this ambiguity and we employed the two terms more carefully in the revised manuscript. Now “regrowth/regrown” refers to growth of a pre-existing neuron’s severed axon and “regeneration/regenerated” refers to newly generated neurons.

3. Not a single experiment here ascertains that increased serotonin concentration in the cord from hyperactive injury-induced ISNs is underlying improved swimming. Authors mention some previous work having established the importance of 5HT release, the lack of receptors does indeed impair recovery, and silencing of Tph1 neurons also impairs recovery, so it is of course a plausible explanation. Yet, the importance of the present work is to claim to have identified the source of 5-HT that supports the recovery: injury-induced ISNs. Therefore, I feel that it is essential to causally link the increased number of ISNs to an actual increase of 5-HT neurotransmission. Figure 3A-I touches upon this question, but there is no hallmark of serotonin concentration in the cord, or release by the neurons. Likewise, synaptic contacts are detected from Tph1-reporter axons (Figure 3I), but no demonstration that these contacts release serotonin is provided. Authors must document this matter experimentally. Currently, the claim that “ISN constantly release serotonin” (line 195, 230, 327, 365, 805 and elsewhere) is not valid. This is all the more important since the origin of these injury-induced ISNs neurons is not clear (see other comment on this). Authors mention they could arise from a neurotransmitter switch. This also leaves open the possibility that ISNs can co-release other transmitters. Hence, authors MUST also firmly exclude that ISNs co-release other neurotransmitters, such as glutamate which is another important activator of swimming CPGs.

Response: We appreciate these important concerns about the concentration of serotonin as well as possible co-transmitters released by regenerated ISNs. We undertook several experiments to answer these questions.

First, we carried out LC-MS experiment to quantify the extracellular concentration of

released serotonin by ISNs in both injury and distal spinal segments (Fig. 3i). The results showed that the concentration of released serotonin in the injury segment was much higher than that measured in the distal segments (Page 10 Line 223-227). This suggests that the regenerated ISNs in the injury spinal segments with their overactive electrophysiological properties and Ca^{2+} oscillation released serotonin constantly and created a microenvironment with a high concentration of serotonin. These results suggest a causal link between the ISN regeneration and significantly increased serotonin levels at the injury site that contrasted with the situation in distal spinal segments away from injury site.

Second, we did anti-Glutamate, anti-GABA and anti-ChAT staining in 1-cut *Tg(tph2:GFP)* fish at four weeks post injury (Supplementary Fig. 2h-j). We quantified the numbers of serotonergic neurons in the injury segments that also co-stained for one of these three markers. We found that about 6% *tph2:GFP*⁺ neurons were positive for anti-glutamate and 6% *tph2:GFP*⁺ neurons were positive for anti-GABA. The numbers of double-stained neurons revealed by anti-Glutamate and anti-GABA labeling were approximately 3 to 4 per injury segment, and almost undetectable for anti-ChAT labeling (Page 10-11 Line 227-231). Considering the low percentage of co-transmitter staining, we think that serotonin is the dominant neurotransmitter released by the regenerated ISNs. We conclude that the dominant neurotransmitter released by regenerated ISNs is serotonin, which facilitates axonal regrowth and locomotor recovery via activation of Htr1b receptors.

4. A large part of the work relies on comparing the ISNs at the lesion site with other 5HT neurons located more distally, the latter serving as controls. This is a rather peculiar strategy in my opinion and non-lesioned animals would have appeared a better option. Since animals do recover whole-body swimming, isn't it possible that distally-located 5HT neurons post-lesion are also influenced by the injury-induced ones or by the global changes in spinal circuits, and may thus be functionally distinct to the ones in non-lesioned fish? Can the authors provide comparisons with non-lesioned fish and clarify this?

Response: We agree with the reviewer that the SCI could induce global changes to some extent in the spinal cord. Such changes could indeed influence the cellular properties of spinal neurons. In the revised manuscript, we attempted to address this point with new experiments using two-photon Ca^{2+} imaging to examine Ca^{2+} homeostasis in ISNs in the spinal segments of uninjured animals and compared it with that in distal spinal segments remote from the injury site. We found that the amplitude and frequency of Ca^{2+} oscillations at the soma and the varicosities of ISNs in spinal segments of uninjured animals and in distal segments of SCI animals were similar and displayed no significant difference (Supplementary Fig. 2g). Importantly, there were no significant differences in the number and amplitude of Ca^{2+} oscillations in ISNs at different spinal segments of the uninjured animals. In contrast, the frequency of Ca^{2+}

oscillations in the soma and the varicosities of the ISNs in the injured segments of the animals were higher than those in distal segments and in spinal segments of uninjured animals (Supplementary Fig. 2g). These findings indicating that the ISNs diverge into different subpopulations described in the revised manuscript (Page 10 Line 212-216).

5. Related to the above, I am confused with the strategy used to claim that ISNs are newly-generated neurons, which is an important finding of the work.

(a) Their number undoubtedly increases at the lesion and significant fraction of them is captured by the EdU injection (Figure S1D). But since the zebrafish at six to eight weeks are still growing (see also minor comment on age), there might be cells in proliferation even in non-lesioned animals. Even if 50% of Tph2:GFP cells incorporated EdU, this does not mean that there is more proliferation than in non-injured animals, and thus that this proliferation is linked to the injury. Authors must report tph2:GFP+/ EdU+ cells in animals without injury and/or at the distal segments where there is presumably no newly-generated ISNs.

Response: We are grateful to the reviewer for raising this point. We performed new experiments to examine the numbers of the tph2:GFP⁺/EdU⁺ cells in uninjured animals and at the distal segments away from the injury site in the injury animals. These results are presented in Supplementary Fig. 1i in the revised manuscript.

Regarding the regeneration of ISNs, we found that compared with a large number of EdU-incorporated ISNs at the injury site, no ISN was labelled by EdU injection in the uninjured animals and very few EdU-incorporated ISNs were seen at the distal spinal segments away from the injury site in the injury animals (see Supplementary Fig. 1i, and Page 7-8 Line 158-160).

As suggested by the reviewer, we also quantified the numbers of ISNs in the animals from one to four months post fertilization in the uninjured animals and present these results in Supplementary Fig. 1c. The results show relatively stable numbers of ISNs at the fish age from one to four months (Page 5 Line 105-106). The lack of EdU labeling and reliable numbers of ISNs prompt us to conclude that in the uninjured animals there is few or even none newly regenerated ISNs.

Finally, similar results to our current findings have been published previously ((McClean and Fetcho, 2004; Montgomery, J., et al., 2018) indicating that the ISNs population in zebrafish achieve morphological and functional maturity as early as 4 days post fertilization.

(b) I didn't get the rationale for looking at EdU incorporation in the 1-cut and 2-cut model (Fig S1D). Where is the sampling made in the 2-cut model, at the first cut or

second cut, or distally? This is not explained/discussed.

Response: We were unclear on this point in the original manuscript and we thank the reviewer for highlighting this. In the revised manuscript, we have added a new illustration with colored markers to better explain how and where the samples were taken and provide bar graph to show the numbers of tph2:GFP⁺/EdU⁺ cell (see Supplementary Fig. 1i).

First, we examined the tph2:GFP⁺/EdU⁺ cells in the injury site of both 1-cut and 2-cut model (Supplementary Fig. 1i, red and orange boxes). The results suggest that the injury-induced ISNs are regenerated, and the regeneration of ISNs is an intraspinal response to SCI.

Second, as suggested by the reviewer, we quantified the tph2:GFP⁺/EdU⁺ cells in the distal segments of both 1-cut and 2-cut model (Supplementary Fig. 1i, box in blue), as well as in uninjured animals (Supplementary Fig. 1i, box in green). We found that none or few ISNs were labeled by EdU injection in the uninjured animals or in distal spinal segments far from the injury site in SCI fish (Page 7-8 Line 158-160). We interpreted these results as indicating that regeneration of ISNs is relatively restricted to the injury site while the number of ISNs in uninjured segments is relatively stable.

6. Related to the above, I would also appreciate some discussion lines regarding the mechanisms that allow ISNs to increase in number (possibly to proliferate) following the injury. Is there a latent source of progenitor cells in the cord, which is engaged into proliferation by lesional tissue? Or is it due a transformation, close to the lesion, of pre-existing cells that would change their neurotransmitter? The latter would make it all the more important to verify co-release of other transmitters.

Response: Done.

First, we discuss the possible origin of these regenerated ISNs and the mechanism responsible for the increase in their number after the injury (see Page 17 Line 372-377).

Second, based on our current results, we do not have enough evidence to suggest a mechanism underlying a neurotransmitter switch. To avoid unnecessary speculation, we have removed this part from the revised manuscript.

For the revision, we did new experiments and found the regenerated ISNs displayed very low percentage of neurons co-labeling with anti-Glutamate, anti-GABA and anti-ChAT staining (Supplementary Fig. 2h-j, Page 10-11 Line 227-231). This suggests that serotonin is the dominant neurotransmitter released by the regenerated

ISNs. In this study, we focus on the function of the regenerated ISNs releasing serotonin constantly that in turn facilitates the axon regrowth of spinal interneurons via activation of Htr1b receptors.

7. I would also appreciate a little bit more background on ISNs. In mammals, there is somewhat of a consensus that the source of serotonin is mostly (exclusively?) supra spinal. Any evidence of intra-spinal serotonin release, either in health or following SCI in mammals too? Also what is known about the projection pattern (e.g., local, ascending, commissural) of ISNs? If one were to discuss the relevance of the findings in rodents, in which the locomotor circuits are more confined to specific segments, do authors think that increasing 5-HT locally at an injury site more rostral, would impact these lumbar circuits? Some discussions along these lines would help to expand the usefulness of the findings.

Response: Done. We have added some discussion on this point that discusses the evidence for 5-HT intraspinal (as opposed to brain-derived) neurons in mammals and the implications for spinal injury in rodents (see Page 17-18 Line 384-395). Our results suggest that increasing 5-HT at the injury site or applying a reasonably specific 5-HT_{1B} agonist would be beneficial in aiding recovery in discrete mammalian spinal injuries.

8. The differential impact of ISNs on excitatory and inhibitory neurons is very interesting. Yet, how this may impact locomotion is not clearly discussed. Both inhibitory and excitatory circuits are required for a proper left-right and inter-segmental coordination of swimming movements, so I'm not sure about the conclusion that "the reestablishment of spinal CPG neural circuits in adult SCI zebrafish is mainly dependent on restoration of excitation mediated by axon regrowth of excitatory glutamatergic interneurons" (line 345). This needs to be clarified and discussed further Also, the control data (Figure S3G) should be incorporated into the main Figure 4F to allow comparison between the lesioned and non-lesioned animals. Also a typo in that figure: Uninjured (instead of uninjury).

Response: As you write, this is a very interesting discussion point. We expanded this section in the revised manuscript. (see Page16, Line 352-360). Spinal transection disrupts the well-organized spinal central pattern generation. The regeneration after SCI in zebrafish revealed in our study clearly shows that axon regrowth of spinal descending excitatory interneurons located in the spinal segments rostral to the injury site contribute to the reconnection of two isolated spinal neural circuits and transmit the necessary excitatory command across the injury site from rostral spinal neural circuit to the caudal one. The regrown long-descending axon of these rostral spinal excitatory interneurons project across both the injury segment and the caudal spinal

segments to possible synapse on the local spinal neurons, which reestablishes the locomotor neural circuit and recovers the rostral-caudal coordination as well as the pattern of locomotor activities. For rostral-caudal coordination, it might mainly rely on the long-descending regrown axon of these spinal excitatory interneurons making synapses on the neurons at each caudal spinal segment that deliver the excitation in a rostral-caudal manner. Left-right coordination could be achieved via activation of local spinal commissure inhibitory interneurons, which directly or indirectly receive the excitation from the axon-regrown spinal excitatory interneurons and inhibit the contralateral side of the neural circuits.

Our control data has been added to Fig. 4f as suggested. We have also changed “uninjury” to “uninjured” in all figures.

9. I didn't find the supplemental video to be convincing at all. It is very short, both fish do a similar number of swimming bouts and the difference in distance travelled is not evident. Also, a scale bar is needed in that video. Please revise it. The data in the Figure 4D is convincing though.

Response: We are sorry for the confusing message given by the video. We have replaced it with a new longer supplementary movie including more than three swimming cycles. The new video compares an uninjured animal, a 2wpi injured animal and a 2wpi injured KO animal. We also added scale bars and illustration showing the head/ tail bending angular differences between uninjured, injured and injured KO animals.

Minor:

1. At multiple times in the manuscript, authors mention that some variables (number of ISNs, swimming behavior, receptor expression levels) at 8 wpi reach the number present before injury. When taking the example of the number of ISNs (line 113), I'm not sure this is what the statistics say in the figure (1B). I see that there is indeed a plateau and a then decrease but the numbers still is significantly higher than pre-injury. Same for the expression of 5-HT receptors (line 269, Figure 4B). Please be more transparent here; that these values stay modulated from the control state does not invalidate the present findings.

Response: We fully agree that the number of ISNs reaches plateau at 8wpi and decreases slower than we expected. It is possible that these ISNs still function to facilitate recovery. We have removed the data regarding the long term 5-HT cell numbers and describe more clearly how ISNs cell numbers change after SCI.

2. Please indicate or represent the sites of dextran injections on histology figures (eg, Figure 1G, K, and others) so that the reader can easily appreciate the directionality of the labelling. Also include the rostro-caudal and D/V axis on all lateral views of the spinal cord (few are missing).

Response: Done. We now indicate the C/R/D/V axes in all the spinal cord images in the revised figures. Concerning the dextran injection sites in the Figure 1G and K, it is difficult for us mark them up since the injection sites were away from the injury sites and it will take large space to show them in the images. We have added illustrations for dextran injection sites in Supplementary Fig. 1e to show how and where we did the injection and how to calculate the axon density. We also explained the injection sites in the Figure legend, the Results section (Page 6 Line 125-129) and the Method section (Page 19-20 Line 430-434).

3. Some questions about age and growth: it is claimed that young adult zebrafish are used, between six to eight weeks and 1.5 to 2 cm in length (line 384). (a) Why was this stage chosen? (b) The authors may want to avoid “adult” because their study model is between 2 stages: juvenile and adult. (c) The authors should specify if the length is the standard length (SL) or the total length (e.g Growth and maturation in the zebrafish, *Danio rerio*: a staging tool for teaching and research, Singleman and Holtzman, 2014, PubMedID : 24979389). (d) Since zebrafish have allometric growth, is it relevant to count tph2+ cells in the same region of 500µm surrounding the lesion site at all the stages?

Response: (a) Previous studies have shown that the spinal neurons and its neural circuit controlling locomotion in zebrafish is fully mature (Ampatzis et al., 2013; Ausborn et al., 2012; Song et al., 2016, 2020) at this stage (around 8 weeks). The serotonergic system is also stable or mature at this age. We are investigating the post-injury modulation of locomotion recovery, specifically ISNs' function. We believe that this age is fine for the current aim.

(b) As the reviewer pointed out, this stage is crossing juvenile and adult stages. We modified our description from “young adults” to “juvenile/adult”.

(c) According to the reference (Singleman and Holtzman, 2014) we added standard length (SL) to our fish body length description (Page 19 Line 411-412).

(d) The data has been added in Supplementary Fig. 1c. The results showed relatively stable numbers of 5-HT neurons in these ages.

4. Maybe say once from the beginning of the result section that this is a complete transection.

Response: Done. We added “complete transection” accordingly (Page 5 Line 98).

5. Showing the acute situation (cord separated) in Figure 1B would help. Likewise,

it is claimed that “when the transected spinal tissues were still separated, no retrogradely labeled interneuron was observed” (line 139). This should be shown and quantified, to ascertain the completeness of the lesion.

Response: Done as requested. First, we added immunohistochemistry images showing *tph2:GFP*⁺ ISNs distribution in *Tg(tph2:GFP)* fish at 1wpi in Supplementary Fig. 1d. The statistic of ISNs' number in 1wpi fish were shown in Fig. 1b. The number of ISNs at the injury sites was decreased at 1 wpi when the spinal cord tissue was still detached. This data combined with the data showing the spinal cord tissue at 1 wpi in Supplementary Fig. 1a verified a complete lesion in our SCI model.

Second, we added a Supplementary Fig. 1f showing the spinal tissue injected with Rhodamine Dextran at 1 wpi, which revealed that no spinal interneurons rostral to the lesion site was retrogradely labeled when the spinal tissue was still detached (Page 6 Line 129-130).

6. line 247: there is no evidence here for a “functional” separation of the subtypes (transcriptomic data).

Response: We changed “functional” to “transcriptomic” (Page 11 Line 249).

7. Figure SIC is missing the situation in control (unlesioned or pre-lesioned) animals. This is needed to claim that the numbers reach control values over time.

Response: We removed this figure since it does not add more information to the current result. Please also see response to Question Minor 1.

8. Figure 2B, 2H: typo at Dextran488.

Response: Corrected.

9. Figure 2, 3 4, 3B, S1, S2D (possibly elsewhere), and associated text: it is incorrect to refer to *tph*⁺ cells. These are *mCherry*⁺/*GFP*⁺ neurons on a reporter fish line.

Response: Corrected. We have modified this in all the figures and associated text.

10. Transcriptomics data (Figure 3K-O): I think that including a broad category of marker genes on the figure panels (e.g. 5-HT transmission, neuronal differentiation,...) would help.

Response: Done. We now added marker category names for each related panel.

11. Line 140: It cannot be claimed that similar results were observed using another

tracer, neurobiotin (Fig S1E) since neurobiotin cells were only counted at 2wpi (while dextran cells were counted at 2, 4 and 8 wpi). I am not sure the neurobiotin data altogether adds much.

Response: These neurobiotin data have been removed from the revised manuscript.

Reviewer #3 (Remarks to the Author):

In this manuscript by Huang et al., the authors investigate the contribution of local serotonergic neurons in the regeneration of the spinal cord – by opposition to central descending neurons – and in locomotor behavior in zebrafish. While the topic covered by this study is overall of high interest, many conclusions are not supported by the data presented in the manuscript and appear overstated.

We appreciate the reviewer's comments and concerns. We did additional experiments to answer reviewer's questions and revised our text to avoid overstating our results. In addition, we added 5-HT_{1B} agonist treatment results in the revised manuscript (Supplementary Fig. 3g). We found that the 5-HT_{1B}-receptor agonist treatment promotes axonal regrowth of spinal interneurons and locomotor recovery (Page 13 Line 289-292).

Major limitations impeding the publication of this manuscript in its current state are listed below.

The first part of this work is designed to provide evidence that spinal cord injury (SCI) induces the regeneration of local serotonergic neurons that are important for locomotor recovery (Figure 1 title “Injury induced regeneration of intraspinal serotonergic neurons (ISNs) is positively correlated with locomotor recovery”). However, the data presented in Figure 1 do not support this conclusion:

a. The optimization of a double spinal transection model is well suited to determine the generation of local serotonergic neurons and their role in motor recovery. However, no clue is provided on the recovery of locomotor patterns in this 2-cut model.

Response: We took the reviewer's suggestion and added the results of forced swimming and free swimming tests in 2-cut SCI animals in the main text. (please see Page 8 Line 166-168). We now show the results of the force swimming and free swimming in 2-cut SCI animals at Fig 2j and k. These data indicate that locomotor function was substantially recovered from 2 wpi to 4 wpi in 2-cut SCI model.

b. No clue is given as to whether the regenerated axons (labelled with rhodamine dextran) are indeed the axons of ISNs, which is the main conclusion provided for this figure (as the title is formulated). Given the results shown in Figure 1K, no colocalization between rhodamine and GFP positive tracts of tph2 neurons can be detected. This rather suggests that axon regeneration and accumulation of tph2 positive cells occurring locally between the two injury sites are two correlated events targeting different cell populations.

Response: On re-reading our first submission, we agree that that this description was unclear and agree that misunderstandings could arise regarding the rhodamine-labeled neuron data. To avoid misunderstanding, we would like to clarify that the ISNs represent intraspinal serotonergic neurons and that these are not spinal interneurons composing the spinal central pattern generator. For the data given in Fig. 1g, Supplementary Fig. 1e and f as well as other related data, we injected rhodamine into the spinal segment caudal to the injury site in order to trace long projecting spinal interneurons whose axon regrow over the injury site after SCI. We did not try to label ISNs with local axon projections at the injury site. We show the morphology of these ISNs' with short axonal projections in Supplementary Fig. 2d.

In order to avoid misunderstandings, we now use “regrowth” to describe growth of a pre-existing neuron's axon and use “regeneration/regenerated” to describe newly regenerated neurons in the revised manuscript. Our principle finding is that regenerated ISNs congregated at the local injury spinal segment and released serotonin to facilitate the axon regrowth of spinal interneuron located in spinal segments rostral to the injury site via Htr1b receptors.

Second, data obtained upon chemogenetic ablation is poorly convincing. While one would expect a large impact of tph2 neurons ablations on locomotor capacity, the amplitude of the effects observed is minimal in the 2-cut injury model (Figure 2J, 2K). Moreover, no control regarding the potential effect of metronidazole on control animals is provided to fully validate the experimental design and demonstrate that metronidazole per se does not affect spinal cord regeneration. Indeed, metronidazole, an antibiotic with well-known anti-inflammatory properties, could act directly on axon regeneration upon SCI and hinder the contribution of local ISNs. This is further supported by earlier reports suggesting that metronidazole interferes with serotonin levels in the nervous systems (Karamanakos PN, Pappas P, Boumba VA, Thomas C, Malamas M, Vougiouklakis T et al. Pharmaceutical agents known to produce disulfiram-like reaction: effects on hepatic ethanol metabolism and brain monoamines. Int. J. Toxicol. 2007; Karamankos, The possibility of serotonin syndrome brought about by the use of metronidazole. Minerva Anesthesiol. 2008).

Response: We are grateful for this additional information about the action of MTZ. We have included this in the discussion (see Page 9 Lines 185). We also did new experiments to answer the reviewer's question.

MTZ induced chemo-genetic ablation is a well-established method widely used in both larval and adult fish for regeneration investigations in the central nervous system (Montgomery et al., 2010; Ohnmacht et al., 2016; Shimizu et al., 2015).

First, we tested for possible effects of MTZ in SCI animals of the *Tg(tph2:Gal4;UAS:mCherry)* transgenic line. MTZ was applied to the fish following

the same protocol as that in the *Tg(tph2:Gal4;UAS:nfsB-mCherry)* transgenic line. MTZ treatment did not kill cells in the animals of *Tg(tph2:Gal4;UAS:mCherry)* transgenic line and allowed us to observe the MTZ effect only. The data show that there was no significant difference with respect to axon regrowth of spinal interneurons or recovery of locomotion between the MTZ treatment group and untreated group (Supplementary Fig. 1k, Page 9 Line 185-189). This suggests that the MTZ treatment itself had no effects on the recovery process of SCI animals.

Second, in our current study, we used MTZ to ablate serotonergic neurons in the animals of *Tg(tph2:Gal4;UAS:nfsB-mCherry)* transgenic line. MTZ killed serotonergic neurons and resulted in reduced serotonin release. Our results show that the released serotonin activated Htr1b receptors and facilitated axon regrowth of spinal interneurons. The decrease in serotonin release by chemo-genetic ablation of serotonergic neurons resulted in reduced recovery in the injured animals of *Tg(tph2:Gal4;UAS:nfsB-mCherry)* transgenic line.

Third, the strategy implemented to identify the molecular signature of local ISNs at the injured spinal segments does not seem valid. Indeed, the authors compare the repertoires of transcripts of GFP+ tph2 neurons depending on their location respective to the injury site, instead of cross-comparing the transcripts enriched against dark cells in each region (i.e. repertoire of GFP+ versus dark in injured segments against the repertoire of GFP+ versus dark in distal segments). Moreover, none of these identified transcripts is validated (using in situ hybridization to compare expression patterns in injured and distal segments).

Response: We agree that the RNA-seq data must be validated. We performed *in situ* hybridization experiments targeting four genes and validated our transcriptomic results as shown in Fig. 3p and Supplementary Fig. 2m (Page 12 Line 254-257). These new data support our conclusion that the regenerated ISNs are functionally distinct subpopulation with the ISNs at distal spinal segments away from injury site.

One aim of the current study was to characterize newly-regenerated ISNs as a molecular and cellular distinct subpopulation of ISNs after spinal cord injury. We considered it best to compare the other ISNs in distal segments remote from the injury site in the same animals under the same conditions.

Fourth, the increased calcium variations in ISNs (in amplitude and frequency, Figure 3H) combined with a co-detection of tph2:GCaMP and SV2 is not sufficient to conclude that “ISNs [...] actively release serotonin that acts on re-growing axons” (title of Figure 3) as no measurement of presynaptic vesicle release probability is provided. Furthermore, there is no specification as to whether Figure 3I represents a single plane or a stack projection, which is critical for the interpretation of this

anatomical argument stating about the apposition of serotonergic varicosities onto rhodamine labelled axons.

Response: We agree with the reviewer that it is difficult to detect presynaptic release of serotonin with current approaches and that serotonin might be released from both extrasynaptic and intrasynaptic varicosities. Extracellular concentration of released serotonin from regenerated ISNs at the injury site could be higher than that from ISNs at the distal segments away from the injury site. Thus, we carried out LC-MS experiment to quantify the concentration of serotonin released extracellularly in the injury and distal segments in the revised manuscript (Fig. 3i). This result provided direct evidence that the extracellular concentration of serotonin in the injury segment is much higher than that in the distal segments (Page 10 Line 223-227). This suggests that the regenerated ISNs displaying overactive electrophysiological properties and Ca^{2+} oscillation, can release serotonin constantly and maintain a microenvironment with a higher concentration of serotonin. We removed the description of intrasynaptic release of serotonin (SV2 data) in the revised manuscript, since such this serotonin could come from both extrasynaptic and intrasynaptic sources.

Finally, the generation of *htr1b 2b* and *7c* mutants is designed to provide evidence on the contribution of local serotonergic signaling on axon regrowth and locomotor recovery. However, both of these phenotypes are not impacted in the two latter mutants (Figure S3E). Moreover, the locomotor phenotype of *htr1b* mutants is minimal (Figure 4D, the distribution of free swimming speeds and max speeds overlap very much) and even undetectable from Video S. Then, what could explain the absence of phenotype in *htr2b* and *7c* mutants while they appear as the most regulated target transcripts after SCI (Figure 4B)? Could gene compensation in the generated mutants hinder phenotypic defects?

Response: The large number of regenerated ISNs releasing high concentration of serotonin at the injury spinal segments appear to play a critical role in promoting axon regrowth of spinal interneurons and locomotor recovery. This prompted us to investigate which 5-HT receptor was involved. We used the prevailing method RNA-seq to uncover the responding subtype. Following bioinformatics analysis, three subtypes (*htr1b*, *htr2b* and *htr7c*) were found to be related to the recovery process after SCI. This analysis is based on the bioinformatics, which indicates only the related genes, not that they are the causal genes. It cannot tell us the functional role of these three subtypes during locomotor recovery.

To further explore the causal link between the receptor subtype and locomotor recovery, we generated *htr1b*, *htr2b* and *htr7c* knock-out animals. We performed SCI to test if the gene mutation could affect locomotor recovery in all three of these knock-out models. As shown in Supplementary Fig. 3f, locomotor recovery after SCI in *htr2b* and *htr7c* knock-out animals was similar to that found in WT fish. However,

the *htr1b* knock-out animals displayed deficiencies in locomotor recovery. We determined that the *htr1b* expression was exclusively up-regulated on the axon regrown spinal excitatory interneurons. Thus, released serotonin presumably activates the *htr1b* receptor on the regrowing axon and facilitates the axon regrowth to enhance locomotor recovery. Finally, pharmacological treatment in the form of application of 5-HT_{1B} agonist further supports our gene mutation experimental results that in the SCI animals the *htr1b* receptor facilitates axon regrowth of spinal interneurons and improves the locomotor recovery (Supplementary Fig. 3g, and Page 13 Line 289-292).

Minor comment:

In general, the manuscript would benefit from text and language editing and from a better description of the experimental paradigms / analysis methods to gain clarity.

Response: Done. We employed a native English-speaker to edit the revised manuscript.

The authors assume in the main text that the targets of ISNs are spinal interneurons. This is stated early in the text (line 167, 185, 193, 221), while evidence for this statement is provided only in Figure 4, which is very misleading.

Response: We agree that our presentation of this point could be improved. We show that serotonin released from the regenerated ISNs facilitate axon regrowth of spinal interneurons as follows.

First, in Fig.1, we showed that the increased number of spinal interneurons with axons regrown across the injury site is associated with increased ISNs number as well as the locomotor recovery.

Second, Fig.2 shows that chemo-genetic ablation of ISNs largely impaired axon regrowth of spinal interneurons as well as the locomotor recovery.

Lastly, Fig.4, demonstrates that the released serotonin from the newly-regenerated ISNs facilitates the axon regrowth of spinal interneurons via activation of Htr1b receptors.

Reviewers' Comments:

Reviewer #1:

Remarks to the Author:

The authors have adequately addressed my previous concerns with additional experiments and changes to the text, and I believe the manuscript is now acceptable for publication.

Reviewer #2:

Remarks to the Author:

In their revised paper, Huang and colleagues have addressed literally all the points I had raised in the original submission. I thus want to congratulate all authors for the pertinence of their responses and the quality and quantity of their revision work. In particular the revised Edu section convincingly demonstrates that proliferation is restricted to the injury site and not due to spontaneous proliferation. The demonstration that distal segments are similar to uninjured animals is also very nice and useful. The addition of serotonin measurement in the bath also adds a lot to the demonstration. On that note, I am still not sure that the work allows to discriminate a higher serotonin concentration due to more ISNs, more hyperactive ISNs, or possibly to both. Nevertheless, this is a remarkable revision work that makes the whole paper now very complete, detailed, and now suited for publication. I only have very minor comments on English below. Although the paper has indeed been substantially improved, I found few phrases that deserve editing.

Figure 3i, please explain what is intensity (cps).

Line 28: what does "these" refer to? Spinal intrinsic mechanisms? But these mechanisms are claimed to be unclear in the above sentence. Please revise that phrase.

Line 37: Spinal cord injury can be abbreviated by SCI, as introduced in the first phrase.

Line 48: twice the word "body".

Line 49: That phrase is not entirely true. It gives the impression that the cause of locomotor dysfunction is only that the injury directly disrupts the CPG network. This is true, but SCI impacts locomotor autonomy by severing the connectivity between brain circuits and the spinal circuits. Obviously, CPG circuits will also undergo functional changes caused by such denervation and inflammatory responses (although the latter may be limited to the injury site). Please rephrase to be more accurate.

Line 82: please define LC-MS or use a broader term at this stage.

Line 88: awkward writing "and result in reorganization...". I don't think ISNs can result in something. The whole pathway does.

Line 92: "target for targeted treatment" is awkward.

Line 113: define Rostro-caudal (R-C) here.

Lines 140-145: awkward writing: the phrase contains twice the fact that the 2-cut is to study ISNs in the absence of descending drive.

Line 219: replace "indicates" by suggests.

Line 223: it says "first, we" but there is no "second". I believe that "second" is line 227 before the staining.

Line 224: please define LC-MS.

Line 227: typo at Glutamate.

Line 358: it is awkward to start a phrase by "while". Use instead "in contrast", or even no transition at all.

Line 359: typo: commissural inhibitory interneurons.

Reviewer #3:

Remarks to the Author:

The authors have adequately addressed most of my comments and concerns in the revised version of the manuscript. The article is now easier to read and to follow.

I acknowledge the authors for their efforts to validate some of their RNAseq candidates to support

the idea that newly generated ISN have a distinct molecular signature.

The issue raised by the intrinsic effect of metronidazole has now been addressed using Tg(*tph2*:GAL4;UAS:mCherry) animals. Dark siblings from Tg(*tph2*:GAL4;UAS:*nfs-B*-mCherry) incrosses would have been an optimal internal control to compare their locomotor capacities to that of *nfs-B* expressing ones, but the option chosen here is sufficiently convincing and covers both locomotion and axon regrowth.

Regarding the potential genetic compensation between 5HT receptors in the different *htr* mutants generated here, this point has not been addressed (no qPCR monitoring the expression level of *htr1b* and *htr2b* transcripts in the *htr7c* mutant for ex.). However, the authors provide a series of evidence, including pharmacology using a selective 5-HT_{1B} agonist, reinforcing the idea that *htr1b* activation does contribute to axon regrowth.

To address the contribution of local axon regrowth induced by the 2-cut model on locomotion, the authors added now the results of forced and free-swimming tests in the main text (lines 166-168). These results suggest that the 2-cut model promotes locomotion recovery, but it is rather a tendency than an increase per se. Please provide the statistics underlying this result. If not significant, please provide a clear interpretation.

Regarding the title of the manuscript, I feel that the present study does not contribute to show any circuit (identified input onto identified target) reorganization per se: I would recommend revising the title. Similarly, I'd recommend editing the following sentence in the abstract (lines 36-37) "In summary, we have found an intraspinal mechanism governing spinal CPG reorganization" as the authors DO NOT DISSECT the reorganization of the CPG circuits here.

The study is now much improved, and I am happy to support the publication of the paper, when the 2 points above will be addressed.

Full responses to each of the points raised are given below.

Reviewer #2 (Remarks to the Author):

In their revised paper, Huang and colleagues have addressed literally all the points I had raised in the original submission. I thus want to congratulate all authors for the pertinence of their responses and the quality and quantity of their revision work. In particular the revised Edu section convincingly demonstrates that proliferation is restricted to the injury site and not due to spontaneous proliferation. The demonstration that distal segments are similar to uninjured animals is also very nice and useful. The addition of serotonin measurement in the bath also adds a lot to the demonstration. On that note, I am still not sure that the work allows to discriminate a higher serotonin concentration due to more ISNs, more hyperactive ISNs, or possibly to both. Nevertheless, this is a remarkable revision work that makes the whole paper now very complete, detailed, and now suited for publication. I only have very minor comments on English below. Although the paper has indeed been substantially improved, I found few phrases that deserve editing.

We thank the reviewer for the comments. We took the reviewer's suggests and meticulously revised our manuscript.

Figure 3i, please explain what is intensity (cps).

Response: We have explained "cps" in the Fig. 3i.

Line 28: what does "these" refer to ? Spinal intrinsic mechanisms? But these mechanisms are claimed to be unclear in the above sentence. Please revise that phrase.

Response: We have revised this sentence to make it more clear (Page 2, Line 27-28, manuscript without tracked changes).

Line 37: Spinal cord injury can be abbreviated by SCI, as introduced in the first phrase.

Response: We have modified this sentence and abbreviated "spinal cord injury" by "SCI" accordingly (Page2, Line 37-38).

Line 48: twice the word "body".

Response: We have rephrased this sentence and removed the word "body" (Page 3, Line 47-48)

Line 49: That phrase is not entirely true. It gives the impression that the cause of locomotor dysfunction is only that the injury directly disrupts the CPG network. This is true, but SCI impacts locomotor autonomy by severing the connectivity between brain circuits and the spinal circuits. Obviously, CPG circuits will also undergo functional changes caused by such denervation and inflammatory

responses (although the latter may be limited to the injury site). Please rephrase to be more accurate.

Response: We have rephrased the sentence to make it more accurate as suggested (Page 3, Line 48-49).

Line 82: please define LC-MS or use a broader term at this stage.

Response: We have defined LC-MS in this sentence as suggested (Page 4, Line 82-83).

Line 88: awkward writing “and result in reorganization...”. I don’t think ISNs can result in something. The whole pathway does.

Response: We have revised this sentence to make it more precise as suggested (Page 4, Line 88).

Line 92 : “target for targeted treatment” is awkward.

Response: We have revised this phrase by removing the word “targeted” (Page 5, Line 92-93).

Line 113: define Rostro-caudal (R-C) here.

Response: We make the definition “rostro-caudal (R-C)” in Line 113 (Page 6) instead of in Line 115 as required now.

Lines 140-145: awkward writing : the phrase contains twice the fact that the 2-cut is to study ISNs in the absence of descending drive.

Response: We have revised this sentence to avoid repeating of information (Page 7, Line 143).

Line 219: replace “indicates” by suggests.

Response: We have replaced “indicates” by “suggests” in this sentence as required (Page 10, Line 216).

Line 223: it says “first, we” but there is no “second”. I believe that “second” is line 227 before the staining.

Response: We have added “Second” in Line 223 (Page10) before “staining” as suggested.

Line 224: please define LC-MS.

Response: We have defined LC-MS in this sentence as required (Page 10, Line 220).

Line 227: typo at Glutamate.

Response: We have corrected the spelling of “Glutamate” (Page 10, Line 224).

Line 358: it is awkward to start a phrase by “while”. Use instead “in contrast”, or even no transition at all.

Response: We have removed the word “while” in this sentence as suggested (Page 16, Line 352).

Line 359: typo: commissural inhibitory interneurons.

Response: We have corrected the phrase “commissural inhibitory interneurons” in this sentence (Page 16, Line 353).

Reviewer #3 (Remarks to the Author):

The authors have adequately addressed most of my comments and concerns in the revised version of the manuscript. The article is now easier to read and to follow.

I acknowledge the authors for their efforts to validate some of their RNAseq candidates to support the idea that newly generated ISN have a distinct molecular signature.

The issue raised by the intrinsic effect of metronidazole has now been addressed using Tg(*tph2*:GAL4;UAS:*mCherry*) animals. Dark siblings from Tg(*tph2*:GAL4;UAS:*nfs-B-mCherry*) incrosses would have been an optimal internal control to compare their locomotor capacities to that of *nfs-B* expressing ones, but the option chosen here is sufficiently convincing and covers both locomotion and axon regrowth.

Regarding the potential genetic compensation between 5HT receptors in the different *htr* mutants generated here, this point has not been addressed (no qPCR monitoring the expression level of *htr1b* and *htr2b* transcripts in the *htr7c* mutant for ex.). However, the authors provide a series of evidence, including pharmacology using a selective 5-HT1B agonist, reinforcing the idea that *htr1b* activation does contribute to axon regrowth.

To address the contribution of local axon regrowth induced by the 2-cut model on locomotion, the authors added now the results of forced and free-swimming tests in the main text (lines 166-168). These results suggest that the 2-cut model promotes locomotion recovery, but it is rather a tendency than an increase per se. Please provide the statistics underlying this result. If not significant, please provide a clear interpretation.

Respond: Thank you for the question. We think there is probably misunderstanding here. The 2-cut model we used in the study is to preclude the influence of brain descending system and testify whether the spinal cord could endogenously repair and recover after SCI. The experiments we performed in the 2-cut model suggested that the animals could not move at all during the first week post 2-cut injury. They were able to swim after one week post 2-cut injury. The locomotor parameters of animals in 2-cut model after 2 weeks and 4 weeks were greatly improved. In 2-cut model, the animals displayed a natural locomotor recovery from the unmovable to flexible locomotion, which supports our idea that independent of supraspinal system there are spinal endogenous mechanisms promoting the repair and recovery of spinal locomotor system after SCI.

We now provide the statistics underlying this result (Page 7, Line 164-166, manuscript without tracked changes). Locomotor parameters did show a tendency to be improved from 2 wpi to 4 wpi in the 2-cut animals. Compared with the animals at

4 wpi post 1-cut model, the animals at 4wpi post 2-cut model displayed similar locomotor parameters. We notice that this differs with the situation of 1-cut model showing that the locomotor parameter becomes better in 4wpi than that in 2wpi. We do not know the exact reason behind the difference and further investigations are needed to address this question.

Regarding the title of the manuscript, I feel that the present study does not contribute to show any circuit (identified input onto identified target) reorganization per se: I would recommend revising the title. Similarly, I'd recommend editing the following sentence in the abstract (lines 36-37) "In summary, we have found an intraspinal mechanism governing spinal CPG reorganization" as the authors DO NOT DISSECT the reorganization of the CPG circuits here.

Respond: We have revised the title and the abstract of the manuscript as suggested.

The study is now much improved, and I am happy to support the publication of the paper, when the 2 points above will be addressed.

We thank the reviewer for the comments.